# Domain Adaptation with Adaptive $f$-Divergence: Tighter Variational Representation and Generalization Bounds

**Zhe Cheng** [1]  **Fode Zhang** [1]  **Yifan Zhu** [1]  **Lingrui Wang** [1]  **Jiaolong Wang** [1]

## Abstract

We study unsupervised domain adaptation (UDA) where measuring cross-domain discrepancy is critical. Most UDA approaches fix a single $f$-divergence a priori, which can be suboptimal across heterogeneous shifts. We propose a framework that (i) tightens the variational lower bound of an $f$-divergence by inserting a learnable, monotone $L$-Lipschitz transform $\tau$ (Tighter-VR), and (ii) selects the divergence family adaptively from data via a likelihood-based criterion. The resulting estimator yields more informative and statistically efficient discrepancy estimates while recovering prior fixed-divergence methods as special cases. Theoretically, we derive a target-risk bound whose three components are a transformed source risk, a Tighter-VR discrepancy between domains, and an ideal-hypothesis residual; we further provide finite-sample guarantees using standard complexity measures. Empirically, on Office-31, Office-Home, Digits, and VisDA-2017, our method consistently improves accuracy over strong baselines, showing that coupling Tighter-VR with adaptive divergence selection is useful for UDA.

## 1. Introduction

UDA addresses the challenge of distribution shift between a labeled *source* domain and an unlabeled *target* domain. In real-world scenarios—e.g., object detection (Li et al., 2023; Nakamura et al., 2024; Liu et al., 2023), sentiment analysis (Ghosal et al., 2020; Dai et al., 2020), and speech recognition (Hu et al., 2024)—annotating data for every new domain is prohibitively expensive. UDA thus aims to align source and target distributions in a common feature space, enabling a model trained on the source to generalize effectively to the target without target labels.

Accurate measurement of distribution discrepancy is critical in UDA. Common approaches include adversarial alignment (minimizing a JS-like divergence via a domain discriminator) (Ganin et al., 2016; Long et al., 2018), optimal transport (Wasserstein distance) (Shen et al., 2018), and moment-matching methods (Long et al., 2015; Sun & Saenko, 2016). More generally, the family of $f$-divergences (e.g., KL, $\chi^2$, Jensen–Shannon) unifies these criteria and underpins theoretical adaptation guarantees. Acuña et al. (Acuna et al., 2021) introduced an $f$-divergence adversarial learning framework ($f$-DAL) that yields a generalization bound over arbitrary $f$-divergences. Wang and Mao (Wang & Mao, 2024) recently refined this theory by proposing a tight $f$-divergence discrepancy ($f$-DD), which removes absolute-value constraints and incorporates a fixed scaling factor to improve convergence rates and error bounds.

Despite these advances, existing UDA methods typically fix the divergence measure *a priori*, applying it uniformly throughout training (Zhang et al., 2019; Shui et al., 2022; Zhang et al., 2020; Nguyen et al., 2022; Tachet des Combes et al., 2020; Mehra et al., 2021). This inflexibility can be suboptimal, since different types of distribution shift—such as covariate shift versus label shift—may be better captured by distinct divergence measures. An adaptive divergence selection mechanism, capable of dynamically adjusting the discrepancy criterion to the data characteristics, would provide more accurate alignment under diverse domain shifts.

We propose an adaptive $f$-divergence framework for unsupervised domain adaptation. Our method introduces two levels of adaptivity: (i) a learnable transformation $\tau$ within the Tighter-VR representation (Birrell et al., 2022), which is optimized jointly with the encoder and classifier to yield a tighter and more flexible discrepancy measure; and (ii) a likelihood-based criterion that adaptively selects the divergence family (e.g., $\alpha$ or Cressie–Read $\beta$) according to data. This unified formulation recovers prior fixed divergences (e.g., $\chi^2$) as special cases (Wang & Mao, 2024; Acuna et al., 2021) while producing more informative and statistically efficient estimates for cross-domain discrepancy.

---

[1]Center of Statistical Research, School of Statistics and Data Science, Southwestern University of Finance and Economics, Chengdu, China. Correspondence to: Fode Zhang <fredzh@swufe.edu.cn>.

*Proceedings of the 43rd International Conference on Machine Learning*, Seoul, South Korea. PMLR 306, 2026. Copyright 2026 by the author(s).

**Contributions.** (i) We derive a Tighter-VR of $f$-divergences by introducing a parametric transform $\tau$, which subsumes classical affine and power transformations and accelerates divergence estimation; we use this representation to derive a target-risk bound (Theorem 3.2) for UDA; (ii) We develop a data-driven mechanism for adaptive divergence selection, using a likelihood-based criterion to adjust the divergence measure to dataset variations; (iii) We provide theoretical guarantees, including generalization bounds derived from Rademacher complexity (Theorems 3.6, 3.7), covering numbers (Theorems 3.10, 3.11), and VC dimension (Theorem 3.12), together with additional insights on how our method extends to data-dependent PAC-Bayesian analysis (Appendix F). (iv) We empirically validate our framework on Office-31, Office-Home, Digits, and VisDA-2017 benchmarks, demonstrating consistent improvements over $f$-DAL (Acuna et al., 2021), $f$-DD (Wang & Mao, 2024), and other strong UDA methods.

## 2. The $f$-Divergence and Tighter-VR

In this section, we briefly recall the definition of the $f$-divergence and its classical Fenchel-duality variational form (Nguyen et al., 2010); we then present a tighter discrepancy measure obtained by introducing a parameterized transform.

Assume two probability measures $\mathbb{P}, \mathbb{Q}$ on a compact space $\mathcal{X}$, both dominated by a $\sigma$-finite measure $\mu$. The $f$-divergence is defined for any closed convex generator $f : (0, \infty) \to \mathbb{R}$, with $f(1) = 0$ and extended-real Fenchel conjugate $f^*$, as

$$\mathcal{D}_f\{\mathbb{P}\|\mathbb{Q}\} = \int_{\mathcal{X}} \frac{d\mathbb{Q}}{d\mu} f\left(\frac{d\mathbb{P}/d\mu}{d\mathbb{Q}/d\mu}\right) d\mu. \qquad (1)$$

Many standard measures—such as Kullback–Leibler (KL), $\chi^2$, and Jensen–Shannon—arise as special cases of $\mathcal{D}_f$. A key property of $f$-divergences is their variational representation via Fenchel-duality. Denote by $f^*$ the Legendre dual of $f$. By convexity $f(u) = \sup_v \{ uv - f^*(v)\}$, which directly yields

$$D_f(\mathbb{P}\|\mathbb{Q}) = \sup_{g \in \mathcal{M}(\mathcal{X})} \left\{ \mathbb{E}_{\mathbb{P}}[g(X)] - \mathbb{E}_{\mathbb{Q}}[f^*(g(X))] \right\}, \quad (2)$$

where $\mathcal{M}(\mathcal{X})$ is the set of real-valued measurable functions on $\mathcal{X}$.

For KL divergence, the Donsker–Varadhan bound takes the form (Ruderman et al., 2012)

$$D_{\mathrm{KL}}(\mathbb{P}\|\mathbb{Q}) = \sup_{g \in \mathcal{M}(\mathcal{X})} \left\{ \mathbb{E}_{\mathbb{P}}[g(X)] - \log \mathbb{E}_{\mathbb{Q}}[e^{g(X)}] \right\}$$
$$\geq \sup_{g \in \mathcal{M}(\mathcal{X})} \left\{ \mathbb{E}_{\mathbb{P}}[g(X)] - \mathbb{E}_{\mathbb{Q}}[f^*(g(X))] \right\},$$

demonstrating that the DV bound is strictly tighter than the generic form (2).

Birrell et al. (Birrell et al., 2022) further tighten this variational lower bound by introducing a family of scalar transforms $\tau : \mathbb{R} \to \mathbb{R}$. Specifically,

$$\mathcal{D}_f\{\mathbb{P}\|\mathbb{Q}\} = \sup_{\substack{g \in \mathcal{M}(\mathcal{X}) \\ \tau \in \mathcal{T}}} \left\{ \mathbb{E}_{\mathbb{P}}[\tau \circ g(X)] \right.$$
$$\left. - \mathbb{E}_{\mathbb{Q}}[f^*(\tau \circ g(X))] \right\},$$

where $\mathcal{T}$ is an appropriate family of transforms. Defining

$$\mathcal{D}_{f,\tau}\{\mathbb{P}\|\mathbb{Q}\} = \sup_{g \in \mathcal{M}(\mathcal{X})} \left\{ \mathbb{E}_{\mathbb{P}}[\tau \circ g(X)] \right.$$
$$\left. - \mathbb{E}_{\mathbb{Q}}[f^*(\tau \circ g(X))] \right\}, \qquad (3)$$

one immediately has $\mathcal{D}_{f,\tau} \leq \mathcal{D}_f$, yielding a Tighter-VR.

For instance, choosing $\tau(x) = a\,x + b$ or $\tau(x) = x^c$ recovers the classical affine and power-law variational bounds, respectively. In the following sections, we show how $\tau$ is instantiated for UDA in practice.

## 3. Unsupervised Domain Adaptation Based on a Tighter Representation

Building on the tighter $f$-divergence in Section 2, we derive a new target-risk bound (Theorem 3.2) that splits the error into three terms: transformed source risk, an $f$-divergence discrepancy, and an ideal-risk residual. We then give finite-sample guarantees for this bound via Rademacher complexity (Theorem 3.6), covering numbers (Theorems 3.10–3.11), and VC dimension (Theorem 3.12). This motivates Section 4, where we make the divergence itself adaptive to data.

### 3.1. Population Risk Bound

Given a labeled source set $S = \{(x_i^s, y_i^s)\}_{i=1}^{n_s} \overset{\text{i.i.d.}}{\sim} \mathbb{P}_s$ and an unlabeled target set $T = \{x_i^t\}_{i=1}^{n_t} \overset{\text{i.i.d.}}{\sim} \mathbb{P}_t$, with labeling functions $f_s$ and $f_t$, UDA seeks a hypothesis $h \in \mathcal{H}$ that predicts on $\mathcal{X}$. A task loss $\ell : \mathcal{Y} \times \mathcal{Y} \to \mathbb{R}^+$ is assumed to be symmetric and satisfies the triangle inequality. We use $R_{\mathbb{P}_s}^\ell(h) = \mathbb{E}_{\mathbb{P}_s}[\ell(h(x), f_s(x))]$ and $R_{\mathbb{P}_t}^\ell(h) = \mathbb{E}_{\mathbb{P}_t}[\ell(h(x), f_t(x))]$ to denote the source and target risks, respectively. Based on Eq. (3), we define the discrepancy measure between the source domain and the target domain as follows

$$\mathcal{D}_{f,\tau}^{h,\mathcal{H}}\{\mathbb{P}_t\|\mathbb{P}_s\} = \sup_{h' \in \mathcal{H}} \left\{ \mathbb{E}_{\mathbb{P}_t}[\tau \circ \ell(h(x), h'(x))] \right.$$
$$\left. - \mathbb{E}_{\mathbb{P}_s}[f^* \circ \tau \circ \ell(h(x), h'(x))] \right\}. \quad (4)$$

*Remark* 3.1. The discrepancy defined in (4) subsumes several existing divergences as special cases. *(i) Location transformation.* When the transformation $\tau$ is restricted

to translations, (4) collapses to the discrepancy introduced by (Agrawal & Horel, 2021). *(ii) Scale transformation.* If $\tau$ is specialised to positive scalings, (4) coincides with the measure proposed by (Wang & Mao, 2024).

Throughout the sequel, we assume that the composition $\tau \circ \ell$ is itself a valid loss. To avoid adding another layer of superscripts, in the theoretical bounds and proofs below we use $R_{\mathbb{P}}^\ell(h)$ as shorthand for $\mathbb{E}_{\mathbb{P}}[\tau \circ \ell(h(x), f(x))]$ whenever the risk is paired with the transformed discrepancy $\mathcal{D}_{f,\tau}$. The definition of the ideal hypothesis $h^*$ and the residual $R^\ell(h^*)$ remain ordinary, untransformed risks as stated. Consequently, the divergences of (Agrawal & Horel, 2021) and (Wang & Mao, 2024) are recovered as proper specialisations of our more general framework.

**Theorem 3.2.** *Let* $h^* = \arg\min\limits_{h \in \mathcal{H}} \left\{ R_{\mathbb{P}_s}^\ell(h) + R_{\mathbb{P}_t}^\ell(h) \right\}$ *be the ideal hypothesis for the ordinary, untransformed source and target risks. Assume that $\tau$ is non-decreasing and $L_1$-Lipschitz continuous. Then for any $h \in \mathcal{H}$ we have*

$$R_{\mathbb{P}_t}^\ell(h) \leq \tilde{R}_{\mathbb{P}_s}^\ell(h) + \mathcal{D}_{f,\tau}^{h,\mathcal{H}}\{\mathbb{P}_t || \mathbb{P}_s\} + L_1 R^\ell(h^*), \quad (5)$$

*where*

$$\tilde{R}_{\mathbb{P}_s}^\ell(h) = \sup_{h' \in \mathcal{H}} \Big\{ \mathbb{E}_{\mathbb{P}_s}\big[ f^* \circ \tau \circ \ell\big(h(x), h'(x)\big)\big] \\ + L_1 \mathbb{E}_{\mathbb{P}_s}\big[\ell\big(h'(x), f_s(x)\big)\big] \Big\},$$

$R^\ell(h^*) := \mathbb{E}_{\mathbb{P}_s}\big[\ell\big(h^*(x), f_s(x)\big)\big] + \mathbb{E}_{\mathbb{P}_t}\big[\ell\big(h^*(x), f_t(x)\big)\big]$ *denotes the total untransformed risk induced by the ideal hypothesis $h^*$.*

*Remark* 3.3 (Interpretation of Theorem 3.2). The monotonicity and Lipschitz assumptions on the transformation $\tau$ are mild and are satisfied by the transform families used in our theoretical analysis. These assumptions are sufficient for the proof, which only requires the implication $\tau(a + b) \leq \tau(a) + L_1|b|$.

Theorem 3.2 bounds the target risk by three quantities:

$$\underbrace{\tilde{R}_{\mathbb{P}_s}^\ell(h)}_{\text{transformed source risk}} + \underbrace{\mathcal{D}_{f,\tau}^{h,\mathcal{H}}\{\mathbb{P}_t || \mathbb{P}_s\}}_{\text{domain discrepancy}} + \underbrace{L_1 R^\ell(h^*)}_{\text{ideal joint risk}}.$$

This structure parallels the upper bounds in (Zhang et al., 2019; Acuna et al., 2021), but with two notable distinctions. First, the source term appears as the *transformed* risk $\tilde{R}_{\mathbb{P}_s}^\ell(h)$ rather than the standard $R_{\mathbb{P}_s}^\ell(h)$, enabling additional control via a suitable choice of $\tau$. Second, the discrepancy $\mathcal{D}_{f,\tau}^{h,\mathcal{H}}\{\mathbb{P}_t || \mathbb{P}_s\}$ explicitly quantifies the distributional gap between the source and target domains.

The residual term $R^\ell(h^*)$ becomes negligible when the hypothesis class $\mathcal{H}$ is sufficiently expressive and the source and target labeling functions are well aligned (Ben-David et al., 2010).

## 3.2. Finite-Sample Generalization Bounds

In practice, the true distributions $\mathbb{P}_s$ and $\mathbb{P}_t$ are unknown—we only have their finite samples $S$ and $T$. Therefore the population terms in Theorem 3.2 must be estimated empirically, and we must control the resulting estimation error. We next derive finite–sample bounds that quantify this gap through standard complexity measures.

**Rademacher complexity bounds** We begin with concentration results expressed in terms of the empirical Rademacher complexity. The focus will encompass both classification and regression issues. Define the labeled and pairwise loss classes

$$\mathcal{G}_s = \{x \mapsto \ell(h(x), f_s(x)) : h \in \mathcal{H}\},$$
$$\mathcal{G}_\Delta = \{x \mapsto \ell(h(x), h'(x)) : h, h' \in \mathcal{H}\}.$$

For compactness, throughout this subsection we write $\mathcal{L}_{\mathcal{H}} := \mathcal{G}_s \cup \mathcal{G}_\Delta$; all complexity, cardinality, covering-number, and VC-dimension terms involving $\mathcal{L}_{\mathcal{H}}$ are taken with respect to this enlarged loss class. We assume the transformed loss values lie in a bounded interval on the considered sample range, so the Lipschitz constants of $\tau$ and $f^*$ are taken on this interval. We first focus on the concentration results of risk $\tilde{R}_{\mathbb{P}_s}^\ell(h)$ and divergence $\mathcal{D}_{f,\tau}^{h,\mathcal{H}}\{\mathbb{P}_t || \mathbb{P}_s\}$.

**Lemma 3.4.** *Assume that the transformation $\tau$ and the Fenchel conjugate $f^*$ are Lipschitz continuous with Lipschitz constants $L_1$ and $L_2$, respectively. Then, with probability at least $1 - \delta$, the following inequality holds for all $h \in \mathcal{H}$*

$$\tilde{R}_{\mathbb{P}_s}^\ell(h) - \tilde{R}_{\hat{\mathbb{P}}_s}^\ell(h) \leq 2L_1(1 + L_2) \widehat{\mathcal{R}}_{n_s}(\mathcal{L}_{\mathcal{H}}, S) \\ + O\Big(\sqrt{\tfrac{\ln(1/\delta)}{n_s}}\Big),$$

*where $\delta > 0$ is a constant, $\widehat{\mathbb{P}}_s$ denotes the empirical measure of the source sample $S$, $\widehat{\mathcal{R}}_{n_s}(\mathcal{L}_{\mathcal{H}}, S)$ indicates the empirical Rademacher complexity of the enlarged loss class $\mathcal{L}_{\mathcal{H}}$ w.r.t. the sample $S$.*

**Lemma 3.5.** *Assume that the assumptions of Lemma 3.4 hold. Then, with probability at least $1 - \delta$, the following inequality holds*

$$\mathcal{D}_{f,\tau}^{h,\mathcal{H}}\{\mathbb{P}_t || \mathbb{P}_s\} - \mathcal{D}_{f,\tau}^{h,\mathcal{H}}\{\widehat{\mathbb{P}}_t || \widehat{\mathbb{P}}_s\} \\ \leq 2L_1 \widehat{\mathcal{R}}_{n_t}(\mathcal{L}_{\mathcal{H}}, T) + 2L_1 L_2 \widehat{\mathcal{R}}_{n_s}(\mathcal{L}_{\mathcal{H}}, S) \\ + O\Big(\sqrt{\tfrac{\log(1/\delta)}{n_s}}\Big) + O\Big(\sqrt{\tfrac{\log(1/\delta)}{n_t}}\Big),$$

*where $\widehat{\mathcal{R}}_{n_s}(\mathcal{L}_{\mathcal{H}}, S)$ and $\widehat{\mathcal{R}}_{n_t}(\mathcal{L}_{\mathcal{H}}, T)$ indicate the empirical Rademacher complexity of the enlarged loss class $\mathcal{L}_{\mathcal{H}}$ w.r.t. the samples $S$ and $T$, respectively.*

Lemma 3.4 therefore gives a standard $O(1/\sqrt{n_s})$ concentration governed by the loss-class complexity. Combining this

result with Lemma 3.5 yields the following finite-sample bound on the target risk.

**Theorem 3.6.** *Assume that the conditions of Theorem 3.2, Lemmas 3.4, and 3.5 hold. Then, with probability at least $1 - \delta$, the target risk can be bounded as follows*

$$
\begin{aligned}
R_{\mathbb{P}_t}^{\ell}(h) \leq{}& \tilde{R}_{\widehat{\mathbb{P}}_s}^{\ell}(h) + 2L_1(1 + 2L_2)\,\widehat{\mathcal{R}}_{n_s}(\mathcal{L}_{\mathcal{H}}, S) \\
&+ 2L_1\,\widehat{\mathcal{R}}_{n_t}(\mathcal{L}_{\mathcal{H}}, T) + \mathcal{D}_{f,\tau}^{h,\mathcal{H}}\big\{\widehat{\mathbb{P}}_t \| \widehat{\mathbb{P}}_s\big\} \\
&+ L_1\,R^{\ell}(h^*) + O\left(\sqrt{\tfrac{\log(1/\delta)}{n_s}} + \sqrt{\tfrac{\log(1/\delta)}{n_t}}\right).
\end{aligned}
\tag{6}
$$

Theorem 3.6 therefore provides a computable upper bound on the target risk in terms of (i) the empirical transformed source risk, (ii) the empirical tighter–VR discrepancy, and (iii) the complexity terms of $\mathcal{L}_{\mathcal{H}}$; the ideal-hypothesis term $R^{\ell}(h^*)$ is data-independent and typically treated as a constant.

The following theorem gives the risk bound when the enlarged loss class has a finite cardinality.

**Theorem 3.7** (Finite-class bound). *Assume that the enlarged loss class $\mathcal{L}_{\mathcal{H}}$ is finite, and define*

$$
B_s = \max_{g \in \mathcal{L}_{\mathcal{H}}}\Big(\sum_{i=1}^{n_s} g(x_i^s)^2\Big)^{1/2},
$$

$$
B_t := \max_{g \in \mathcal{L}_{\mathcal{H}}}\Big(\sum_{i=1}^{n_t} g(x_i^t)^2\Big)^{1/2}.
$$

*Then, with probability at least $1 - \delta$, we have*

$$
\begin{aligned}
R_{\mathbb{P}_t}^{\ell}(h) \leq{}& \tilde{R}_{\widehat{\mathbb{P}}_s}^{\ell}(h) + 2L_1\bigg[(1 + 2L_2)\frac{B_s\sqrt{2\ln|\mathcal{L}_{\mathcal{H}}|}}{n_s} \\
&+ \frac{B_t\sqrt{2\ln|\mathcal{L}_{\mathcal{H}}|}}{n_t}\bigg] + \mathcal{D}_{f,\tau}^{h,\mathcal{H}}(\widehat{\mathbb{P}}_t\|\widehat{\mathbb{P}}_s) \\
&+ L_1\,R^{\ell}(h^*) + O\left(\sqrt{\tfrac{\ln(1/\delta)}{n_s}} + \sqrt{\tfrac{\ln(1/\delta)}{n_t}}\right).
\end{aligned}
\tag{7}
$$

Theorem 3.7 reveals that, beyond the transformed source risk and the domain discrepancy, the finite–sample contribution to the target risk decays at rates $O\big(\sqrt{\ln|\mathcal{L}_{\mathcal{H}}|/n_s}\big)$ and $O\big(\sqrt{\ln|\mathcal{L}_{\mathcal{H}}|/n_t}\big)$, highlighting the dual role of sample size and model cardinality.

We observe that the bounds of Theorems 3.6 and 3.7 depend on the complexity or cardinality of the enlarged loss class $\mathcal{L}_{\mathcal{H}}$ rather than the hypothesis space $\mathcal{H}$ itself. In the binary 0–1 case, the labeled part reduces to the complexity of $\mathcal{H}$, while the pairwise part is captured by the disagreement class.

**Corollary 3.8** (Binary case, 0–1 loss). *Let $h \in \mathcal{H}$ be the hypotheses taking the values in $\{+1, -1\}$, and let the loss function $\ell$ be the zero-one loss. Define $\mathcal{H}_{\Delta} := \{x \mapsto \mathbb{1}_{h(x) \neq h'(x)} : h, h' \in \mathcal{H}\}$. Assume that the assumptions of Theorem 3.6 hold. Then, with probability at least $1 - \delta$, we have*

$$
\begin{aligned}
R_{\mathbb{P}_t}^{\ell}(h) \leq{}& \tilde{R}_{\widehat{\mathbb{P}}_s}^{\ell}(h) + L_1\,\widehat{\mathcal{R}}_{n_s}(\mathcal{H}, S) \\
&+ 4L_1L_2\,\widehat{\mathcal{R}}_{n_s}(\mathcal{H}_{\Delta}, S) + 2L_1\,\widehat{\mathcal{R}}_{n_t}(\mathcal{H}_{\Delta}, T) \\
&+ \mathcal{D}_{f,\tau}^{h,\mathcal{H}}\big\{\widehat{\mathbb{P}}_t\|\widehat{\mathbb{P}}_s\big\} + L_1\,R^{\ell}(h^*) \\
&+ O\left(\sqrt{\tfrac{\log(1/\delta)}{n_s}} + \sqrt{\tfrac{\log(1/\delta)}{n_t}}\right).
\end{aligned}
$$

*Remark* 3.9. The bound in Corollary 3.8 separates the labeled loss class and the pairwise disagreement class. The labeled 0–1 loss contributes the complexity of $\mathcal{H}$, while the discrepancy terms contribute the complexity of $\mathcal{H}_{\Delta}$.

**Covering-number bounds (Zhang, 2002).** When the enlarged loss class $\mathcal{L}_{\mathcal{H}}$ has finite metric entropy, the target risk can also be controlled via $\epsilon$-covering numbers, sometimes yielding tighter guarantees than Rademacher complexity. For a pseudo-metric $\rho$, let $\mathcal{C}(\mathcal{L}_{\mathcal{H}}, \epsilon, \rho)$ denote the smallest number of $\epsilon$-balls (in $\rho$) required to cover $\mathcal{L}_{\mathcal{H}}$. For any $g \in \mathcal{L}_{\mathcal{H}}$ and $p > 0$ we write $\|g\|_{p,S} = \big(\frac{1}{n_s}\sum_{i=1}^{n_s}|g(x_i^s)|^p\big)^{1/p}$, $\|g\|_{p,T} = \big(\frac{1}{n_t}\sum_{j=1}^{n_t}|g(x_j^t)|^p\big)^{1/p}$. We are now ready to state the covering-number generalization bound.

**Theorem 3.10.** *Assume that the assumptions of Theorem 3.6 hold and there exists a constant $c > 0$ such that $\|g\|_{2,S} \leq c$ and $\|g\|_{2,T} \leq c$ for all $g \in \mathcal{L}_{\mathcal{H}}$. Then, with probability at least $1 - \delta$, we have*

$$
\begin{aligned}
R_{\mathbb{P}_t}^{\ell}(h) \leq{}& \tilde{R}_{\widehat{\mathbb{P}}_s}^{\ell}(h) + L_1 R^{\ell}(h^*) \\
&+ 2L_1\bigg[(1 + 2L_2)\inf_{\varepsilon > 0}\Big(\varepsilon + \tfrac{c\sqrt{2}}{\sqrt{n_s}}\sqrt{\ln\mathcal{C}(\mathcal{L}_{\mathcal{H}}, \varepsilon, \|\cdot\|_{1,S})}\Big) \\
&\quad + \inf_{\varepsilon > 0}\Big(\varepsilon + \tfrac{c\sqrt{2}}{\sqrt{n_t}}\sqrt{\ln\mathcal{C}(\mathcal{L}_{\mathcal{H}}, \varepsilon, \|\cdot\|_{1,T})}\Big)\bigg] \\
&+ \mathcal{D}_{f,\tau}^{h,\mathcal{H}}(\widehat{\mathbb{P}}_t\|\widehat{\mathbb{P}}_s) + O\left(\sqrt{\tfrac{\log(1/\delta)}{n_s}} + \sqrt{\tfrac{\log(1/\delta)}{n_t}}\right).
\end{aligned}
\tag{8}
$$

Theorem 3.10 gives the upper bound using a fixed level of granularity $\varepsilon$, taking the infimum over $\varepsilon$. The following theorem, based on the Dudley chaining technique (Dudley, 1967; Mendelson & Vershynin, 2003), gives the upper bound of the target risk by integrating over different levels of granularity.

**Theorem 3.11.** *Assume that the assumptions of Theo-*

*rem 3.10 hold. Let*

$$\mathcal{C}_s(\eta) := \mathcal{C}(\mathcal{L}_\mathcal{H}, \eta, \|\cdot\|_{2,S}),$$
$$\mathcal{C}_t(\eta) := \mathcal{C}(\mathcal{L}_\mathcal{H}, \eta, \|\cdot\|_{2,T}),$$

*and define*

$$\Psi_s := \inf_{\varepsilon \in [0,c/2]} \left( 4\varepsilon + \frac{12}{\sqrt{n_s}} \int_\varepsilon^{c/2} \sqrt{\ln \mathcal{C}_s(\eta)}\, d\eta \right),$$

$$\Psi_t := \inf_{\varepsilon \in [0,c/2]} \left( 4\varepsilon + \frac{12}{\sqrt{n_t}} \int_\varepsilon^{c/2} \sqrt{\ln \mathcal{C}_t(\eta)}\, d\eta \right).$$

*Then, with probability at least $1 - \delta$, we have*

$$R_{\mathbb{P}_t}^\ell(h) \leq \tilde{R}_{\widehat{\mathbb{P}_s}}^\ell(h) + \mathcal{D}_{f,\tau}^{h,\mathcal{H}}\{\widehat{\mathbb{P}}_t \| \widehat{\mathbb{P}}_s\} + L_1\, R^\ell(h^*)$$
$$+ 2L_1\big((1 + 2L_2)\Psi_s + \Psi_t\big)$$
$$+ O\left( \sqrt{\frac{\ln(1/\delta)}{n_s}} + \sqrt{\frac{\ln(1/\delta)}{n_t}} \right).$$

Theorem 3.11 provides an upper bound on the target risk via an entropy integral over the scale $\varepsilon$. Since the covering numbers $\mathcal{C}(\mathcal{L}_\mathcal{H}, \varepsilon, \|\cdot\|_{2,T})$ and $\mathcal{C}(\mathcal{L}_\mathcal{H}, \varepsilon, \|\cdot\|_{2,S})$ are non-increasing in $\varepsilon$, Dudley's entropy integral connects the rate at which these covering numbers decay with the resulting complexity term.

**VC-dimension bound**   We restrict to binary classification with the 0–1 loss, where the hypothesis space admits a finite VC-dimension. Under this setting, we obtain the following combinatorial generalization bound.

**Theorem 3.12** (VC bound). *Let $VC(\mathcal{L}_\mathcal{H}, T)$ and $VC(\mathcal{L}_\mathcal{H}, S)$ be the VC-dimensions of the enlarged loss class $\mathcal{L}_\mathcal{H}$ associated with samples $T$ and $S$, respectively. We assume that the conditions of Theorem 3.6 hold, $n_t \geq VC(\mathcal{L}_\mathcal{H}, T)$ and $n_s \geq VC(\mathcal{L}_\mathcal{H}, S)$. Then, with probability at least $1 - \delta$, we have*

$$R_{\mathbb{P}_t}^\ell(h) \leq \tilde{R}_{\widehat{\mathbb{P}_s}}^\ell(h) + L_1\, R^\ell(h^*)$$
$$+ 2L_1 \left[ (1 + 2L_2) \sqrt{\frac{2\, VC(\mathcal{L}_\mathcal{H}, S)\, \log\big(\frac{e\, n_s}{VC(\mathcal{L}_\mathcal{H}, S)}\big)}{n_s}} \right.$$
$$\left. + \sqrt{\frac{2\, VC(\mathcal{L}_\mathcal{H}, T)\, \log\big(\frac{e\, n_t}{VC(\mathcal{L}_\mathcal{H}, T)}\big)}{n_t}} \right]$$
$$+ \mathcal{D}_{f,\tau}^{h,\mathcal{H}}(\widehat{\mathbb{P}}_t \| \widehat{\mathbb{P}}_s) + O\left( \sqrt{\frac{\log(1/\delta)}{n_s}} + \sqrt{\frac{\log(1/\delta)}{n_t}} \right).$$

*Remark* 3.13. Compared with the Rademacher and covering-number bounds, the VC bound replaces the empirical complexity terms by a combinatorial term governed by $VC(\mathcal{L}_\mathcal{H})$ and the sample sizes; this term is distribution-free and independent of the realized sample values. Moreover, our analysis is extended to the multi-class case, with the corresponding derivation provided in Appendix E.9.

*Remark* 3.14 (Connection to Fast-Rate Bounds). Our decomposition–empirical (transformed) source loss plus an $f$-divergence discrepancy plus an irreducible ideal-hypothesis term–has the same structure as fast-rate PAC-Bayesian (Alquier, 2024; Seldin et al., 2012; Tolstikhin & Seldin, 2013; Yang et al., 2019) and information-theoretic generalization bounds (Hellström & Durisi, 2022; Wu et al., 2024; Wang & Mao, 2023; Xu & Raginsky, 2017). In particular, refined localized or data-dependent PAC-Bayes techniques can tighten the discrepancy and complexity terms without changing our core algorithm. Further discussion and supporting details regarding the PAC-Bayes analysis are provided in Appendix F.

Taken together, Theorems 3.6, 3.7, 3.10, 3.11, and 3.12 show that the finite–sample error can be controlled via Rademacher complexity, metric entropy, or VC-dimension, allowing the analyst to choose the sharpest bound for a given hypothesis class. Proofs for these theorems and lemmas appear in Appendix E.

# 4. Adaptive Selection of $f$-Divergence

Choosing an appropriate discrepancy is important in both classical statistics (Ali & Silvey, 1966) and modern machine learning (Shalev-Shwartz & Ben-David, 2014), yet practical selection often defaults to cross-validation or heuristics. A large share of commonly used $f$-divergences is captured by *parameterized* families—e.g., $\alpha$-divergences, Cressie–Read power divergences, and $\gamma$-divergences—which cover KL, Pearson $\chi^2$, Hellinger, and Itakura–Saito-type limits (see Appendix C). Our approach is to *adapt* the discrepancy by estimating the family parameter(s) via a likelihood-based criterion, thereby learning the most suitable $f$-divergence from data rather than fixing it a priori.

In what follows, we present the full construction using the Cressie–Read power-divergence family, indexed by $\beta$, as a running example; the same selection principle extends straightforwardly to other parameterized $f$-divergence families, including the $\alpha$-family (see Appendix C.1). To obtain a tractable selection rule, we estimate $\beta$ with a Tweedie-linked likelihood score and then insert the selected generator into the Tighter-VR objective.

**Likelihood-based selection via a $\beta$-linked likelihood**   We instantiate the adaptive scheme by selecting the Cressie–Read parameter $\beta$ via a Tweedie-linked likelihood score: For notational brevity, $x^s$ and $x^t$ in Eqs. (9)–(11) denote the positive scalar scores used by the selection model, rather

than raw images or vector-valued features.

$$p_{\text{sel}}(x^s, x^t, \beta, \phi) = \frac{1}{z(x^s, \beta, \phi)} \exp\left\{k(x^t, \beta)\right.$$
$$\left. - \frac{1}{\phi} \Delta_\beta(x^t \| x^s)\right\}. \quad (9)$$

where $k(x^t, \beta) = \frac{\beta-1}{2} \ln x^t$, $z(x^s, \beta, \phi)$ is a normalizing constant (see Appendix E.10), and

$$\Delta_\beta(x^t \| x^s) = \frac{(x^t)^\beta + (\beta-1)(x^s)^\beta - \beta\, x^t (x^s)^{\beta-1}}{\beta(\beta-1)}$$

is the scalar Tweedie deviance used only for parameter selection. The selected discrepancy used in the variational objective is the Cressie–Read $f$-divergence with generator

$$f_\beta(t) = \frac{t^\beta - \beta t + \beta - 1}{\beta(\beta-1)}, \qquad \beta \neq 0, 1,$$

namely

$$D_\beta(\mathbb{P}_t \| \mathbb{P}_s) = \int_{\mathcal{X}} p_s(x)\, f_\beta\left(\frac{p_t(x)}{p_s(x)}\right) \mathrm{d}\mu(x)$$
$$= \frac{1}{\beta(\beta-1)} \int_{\mathcal{X}} \left[p_t(x)^\beta p_s(x)^{1-\beta} - \beta\, p_t(x)\right.$$
$$\left. + (\beta-1)\, p_s(x)\right] \mathrm{d}\mu(x). \quad (10)$$

This is a standard Csiszár $f$-divergence family; in particular, $\beta = 2$ gives one half of the Pearson $\chi^2$ divergence. The optimal parameter $\beta^*$ can be selected by maximizing the joint likelihood score on paired scalar score batches $\{(x_i^s, x_i^t)\}_{i=1}^m$, formed within synchronized mini-batches with $m \leq \min\{n_s, n_t\}$. That is

$$\beta^* = \arg\max_\beta \max_\phi \sum_{i=1}^m \left[k(x_i^t, \beta)\right.$$
$$\left. - \frac{1}{\phi} \Delta_\beta(x_i^t \| x_i^s) - \ln z(x_i^s, \beta, \phi)\right] \quad (11)$$

Substituting the optimal $\beta^*$ in Eq. (10) gives the selected Cressie–Read divergence.

**Extension to the $\alpha$-divergence family.** The same selection principle can also be applied to the $\alpha$-divergence family. The $\alpha$-family and the Cressie–Read $\beta$-family are known to be connected through suitable nonlinear transformations and escort-measure constructions (Amari & Cichocki, 2010; Cichocki & Amari, 2010), but our method does not require a global sample-level bijection between them. One may instead choose an $\alpha$-family generator $f_\alpha$, estimate $\alpha$ by an analogous likelihood-based or validation-based criterion, and substitute the selected generator into the same Tighter-VR objective. In our experiments, we instantiate this idea with the Cressie–Read $\beta$-family because it provides a convenient selection score and recovers the Pearson $\chi^2$ point at $\beta = 2$.

**Connection to the tighter-VR discrepancy.** For the Cressie–Read instantiation, the data–driven procedure produces an optimal parameter $\beta^*$, which determines the selected parameterized $f$-divergence; an analogous $\alpha$-family choice would replace $f_{\beta^*}$ with the selected $f_{\alpha^*}$. For the Cressie–Read family, the conjugate $f_\beta^*(v)$ is evaluated only on a compact interval contained in its effective domain $1 + (\beta - 1)v > 0$. Thus the admissible transforms $\tau$ are restricted so that $v = \tau(\ell)$ remains in this interval on the considered sample range; this is the same interval on which the local Lipschitz constant $L_2$ is taken. Substituting the selected Cressie–Read divergence into Eq. (4), the discrepancy term in our UDA objective becomes

$$\mathcal{D}_{f_{\beta^*}, \tau}^{h, \mathcal{H}}\{\mathbb{P}_t \| \mathbb{P}_s\} = \sup_{h' \in \mathcal{H}} \left\{\mathbb{E}_{x \sim \mathbb{P}_t}\left[\tau(\ell(h(x), h'(x)))\right]\right.$$
$$\left. - \mathbb{E}_{x \sim \mathbb{P}_s}\left[f_{\beta^*}^*\left(\tau(\ell(h(x), h'(x)))\right)\right]\right\}. \quad (12)$$

Hence, *divergence selection and representation alignment are now jointly adaptive*: the parameter $\tau$ is learned during training, while the choice of $f$ is updated via the likelihood criterion, yielding an end-to-end objective that tailors both the divergence family and its variational lower bound to the observed source–target pair.

## 5. Training Algorithm and Experiments

### 5.1. Learning Algorithms

Our network follows the encoder–classifier split common in UDA. Let $h_{\text{rep}} : \mathcal{X} \to \mathcal{Z}$ be a feature encoder and $h_{\text{cls}} : \mathcal{Z} \to \mathcal{Y}$ a label head; the forward hypothesis is $h = h_{\text{cls}} \circ h_{\text{rep}}$. During training we instantiate a *critic* $h' = h_{\text{cls}}' \circ h_{\text{rep}}$ that shares the encoder but has independent classifier weights, as in adversarial discrepancy methods. **Fig. 2** illustrates the overall architecture of our model. For further implementation details, please refer to Appendix G.

**Objective.** Given labelled source mini-batch $\hat{\mu} = \{(x_i^s, y_i^s)\}_{i=1}^{m_s}$ and unlabelled target mini-batch $\hat{\nu} = \{x_j^t\}_{j=1}^{m_t}$, we solve the saddle objective

$$\min_h \max_{h'} \underbrace{\mathbb{E}_{\hat{\mu}}\left[\tau(\ell(h(x), y))\right]}_{\text{transformed source cls.}}$$
$$+ \eta \left(\underbrace{\mathbb{E}_{\hat{\nu}}\left[\tau(\ell(h, h'))\right] - \mathbb{E}_{\hat{\mu}}\left[f^*\left(\tau(\ell(h, h'))\right)\right]}_{\text{tighter-VR discrepancy } \tilde{d}_{\hat{\mu}, \hat{\nu}}^\tau(h, h')}\right) \quad (13)$$

where $\ell$ is the empirical training surrogate loss, $f^*$ is the Fenchel conjugate of the chosen $f$-divergence, and $\eta > 0$ balances accuracy and alignment. The theoretical risk bounds above are stated for metric task losses, while this surrogate is used for optimization in the empirical model. For the adaptive variant, the Tighter-VR discrepancy term is replaced with the empirical analogue of Eq. (12).

**Learnable transform $\tau$.** We parameterize $\tau$ as a simple, monotone, $L_1$-Lipschitz map (defaulting to an affine form, see Appendix G for additional families) whose scalar parameters are initialized to $(a, b) = (1, 0)$ and learned jointly with the network parameters. This re-scaling stabilizes training and improves adaptation for both $\chi^2$ and other $f$-divergences.

**Optimization.** We alternate one gradient step on $(h_{\mathrm{rep}}, h_{\mathrm{cls}}, a, b)$ (*minimisation*) and one on $h'_{\mathrm{cls}}$ (*maximisation*). Standard tricks—gradient reversal for the critic, weight decay, and learning-rate warm-up—are applied exactly as in prior work (Acuna et al., 2021; Zhang et al., 2019).

### 5.2. Experiments

We empirically evaluate our adaptive $f$-divergence UDA framework with five goals: (i) compare against strong methods on Office-31, Office-Home, VisDA-2017, and Digits; (ii) quantify the effect of the tighter discrepancy and learnable transformation through ablations; (iii) analyze adaptive divergence dynamics and learned feature representations; (iv) test the method with a ViT backbone and report significance analyses (Appendix I.4); and (v) examine an RK2 optimizer variant (Appendix I.5). Additional divergence-selection and transform ablations are provided in Appendices I.2 and I.1.

**Experimental Setup** We evaluate on four standard UDA suites: **Office-31** (Saenko et al., 2010) (31 classes, A→W, D→W, . . . ), **Office-Home** (Venkateswara et al., 2017) (65 classes, 12 domain pairs), **VisDA-2017** (Peng et al., 2017) (12 classes, synthetic → real) and the **Digits** task MNIST ↔ USPS (Hull, 1994). Target-domain top-1 accuracy is the primary metric for all reported comparisons. For Office datasets we use an ImageNet-pretrained ResNet-50 encoder; for VisDA-2017 we use a pretrained ResNet-101 encoder; and for Digits we adopt LeNet (LeCun et al., 1998). For the Office benchmarks, Tables 1 and 2 compare against *source-only* ResNet-50 (He et al., 2016), DANN (Ganin et al., 2016), JAN (Long et al., 2017), CDAN (Long et al., 2018), MDD (Zhang et al., 2019), $f$-DAL (Acuna et al., 2021), and $f$-DD (Wang & Mao, 2024); Office-31 additionally includes GTA (Sankaranarayanan et al., 2018) and MCD (Saito et al., 2018). The detailed experimental setup and hyperparameters can be found in Appendix H.1. The appendix further includes comparisons with optimal-transport baselines (Appendix I.3), DeiT-S/16 results on DomainNet C/P/R/S (Appendix I.4.2), broader recent UDA reference comparisons (Appendix I.4.3), and adaptive-$\beta$ results on Office-Home and VisDA-2017 (Appendix I.6). These results complement the main tables and show that the proposed discrepancy remains competitive beyond the original ResNet-based evaluation protocol.

**Overall Benchmark Results.** Tables 1–3 report the target-domain accuracies. For $f$-divergence methods, the main fixed-divergence comparison uses Pearson $\chi^2$ unless otherwise stated. On **Office-31** our model attains **90.4%** average, surpassing $f$-DAL (89.5%) and $f$-DD (89.9%); the largest gains appear on the challenging shifts A→D, D→A, and W→A (+0.8 pp over $f$-DD). On the harder **Office-Home** benchmark we reach **69.9%** versus 68.5% ($f$-DAL) and 69.2% ($f$-DD), winning 11 / 12 pairs. On the lightweight **Digits** task our affine model achieves **96.9%** mean accuracy, edging past $f$-DD (96.4%) and $f$-DAL (96.3%). Finally, on the large-scale **VisDA-2017** benchmark, our method reaches 78.9% accuracy, with a clear margin over $f$-DAL (74.3%) and $f$-DD (76.2%). These consistent improvements indicate that tightening the variational bound and learning the transform $\tau$ provide an accuracy gain even when the underlying divergence family is fixed.

**Ablation Study: Tighter discrepancy measure with different Transformation.** To disentangle the effect of the tight bound and the choice of transformation $\tau$, we compare six variants (Table 4):

(i) $f$-DAL (loose VR, no transform), (ii) $f$-DD (tight VR, fixed $\tau$), and four Tighter-VR models with learnable *Power*, *Exponential*, *Sigmoid*, or *Affine* maps. Across the three benchmarks the affine variant delivers the largest gain, e.g. +0.5 pp on Office-31 and +0.5 pp on Digits relative to the static identity, showing the benefit of a *learnable* re-scaling. Other nonlinear maps provide smaller, task-dependent improvements over the static baseline, while the original loose VR ($f$-DAL) lags further behind, indicating that both Tighter-VR and a learnable transformation contribute to the final accuracy.

**Adaptive Cressie–Read $\beta$ Selection Dynamics.** To assess our framework's ability to *learn* the optimal divergence, we conduct two experiments on the Digits benchmarks. **Grid search:** train with fixed $\beta \in \{0.1, 0.3, \ldots, 5.0\}$ and record peak target accuracy for each pair. **Joint tuning:** (i) fully adaptive–$\beta$, where $\beta$ is updated by gradient ascent on the likelihood score each epoch; (ii) "freeze" ablation, where $\beta$ is learned for the first 10 epochs then held constant.

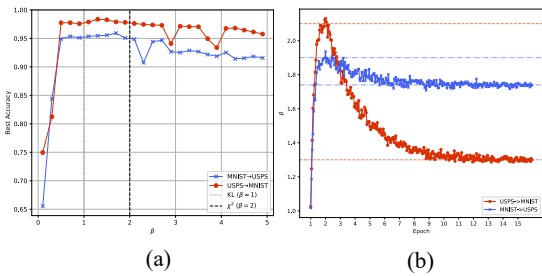

(a)           (b)

*Figure 1.* Adaptive Cressie–Read $\beta$ selection dynamics on Digits.

*Table 1.* Accuracy (%) on Office-31 (mean $\pm$ std).

| Method | A $\rightarrow$ W | D $\rightarrow$ W | W $\rightarrow$ D | A $\rightarrow$ D | D $\rightarrow$ A | W $\rightarrow$ A | Avg |
|---|---|---|---|---|---|---|---|
| ResNet-50 | $68.4 \pm 0.2$ | $96.7 \pm 0.1$ | $99.3 \pm 0.1$ | $68.9 \pm 0.2$ | $62.5 \pm 0.3$ | $60.7 \pm 0.3$ | 76.1 |
| DANN | $82.0 \pm 0.4$ | $96.9 \pm 0.2$ | $99.1 \pm 0.1$ | $79.7 \pm 0.4$ | $68.2 \pm 0.4$ | $67.4 \pm 0.5$ | 82.2 |
| JAN | $85.4 \pm 0.3$ | $97.4 \pm 0.2$ | $99.8 \pm 0.2$ | $84.7 \pm 0.3$ | $68.6 \pm 0.3$ | $70.0 \pm 0.4$ | 84.3 |
| GTA | $89.5 \pm 0.5$ | $97.9 \pm 0.3$ | $99.8 \pm 0.4$ | $87.7 \pm 0.5$ | $72.8 \pm 0.3$ | $71.4 \pm 0.4$ | 86.5 |
| MCD | $88.6 \pm 0.2$ | $98.5 \pm 0.1$ | $\mathbf{100.0 \pm 0.0}$ | $92.2 \pm 0.2$ | $69.5 \pm 0.1$ | $69.7 \pm 0.3$ | 86.5 |
| CDAN | $94.1 \pm 0.1$ | $98.6 \pm 0.1$ | $\mathbf{100.0 \pm 0.0}$ | $92.9 \pm 0.2$ | $71.0 \pm 0.3$ | $69.3 \pm 0.3$ | 87.7 |
| MDD | $94.5 \pm 0.3$ | $98.4 \pm 0.1$ | $\mathbf{100.0 \pm 0.0}$ | $93.5 \pm 0.2$ | $74.6 \pm 0.3$ | $72.2 \pm 0.1$ | 88.9 |
| $f$-DAL | $95.4 \pm 0.7$ | $98.8 \pm 0.1$ | $\mathbf{100.0 \pm 0.0}$ | $93.8 \pm 0.4$ | $74.9 \pm 1.5$ | $74.2 \pm 0.5$ | 89.5 |
| $f$-DD($\chi^2$-DD) | $95.3 \pm 0.2$ | $98.7 \pm 0.1$ | $\mathbf{100.0 \pm 0.0}$ | $95.0 \pm 0.4$ | $74.7 \pm 0.5$ | $75.6 \pm 0.2$ | 89.9 |
| Ours (Affine) | $\mathbf{95.5 \pm 0.3}$ | $\mathbf{98.9 \pm 0.1}$ | $\mathbf{100.0 \pm 0.0}$ | $\mathbf{95.8 \pm 0.2}$ | $\mathbf{75.5 \pm 0.5}$ | $\mathbf{76.4 \pm 0.2}$ | $\mathbf{90.4}$ |

*Table 2.* Accuracy (%) on the Office-Home benchmark.

| Method | Ar $\rightarrow$ Cl | Ar $\rightarrow$ Pr | Ar $\rightarrow$ Rw | Cl $\rightarrow$ Ar | Cl $\rightarrow$ Pr | Cl $\rightarrow$ Rw | Pr $\rightarrow$ Ar | Pr $\rightarrow$ Cl | Pr $\rightarrow$ Rw | Rw $\rightarrow$ Ar | Rw $\rightarrow$ Cl | Rw $\rightarrow$ Pr | Avg |
|---|---|---|---|---|---|---|---|---|---|---|---|---|---|
| ResNet-50 | 34.9 | 50.0 | 58.0 | 37.4 | 41.9 | 46.2 | 38.5 | 31.2 | 60.4 | 53.9 | 41.2 | 59.9 | 46.1 |
| DANN | 45.6 | 59.3 | 70.1 | 47.0 | 58.5 | 60.9 | 46.1 | 43.7 | 68.5 | 63.2 | 51.8 | 76.8 | 57.6 |
| JAN | 45.9 | 61.2 | 68.9 | 50.4 | 59.7 | 61.0 | 45.8 | 43.4 | 70.3 | 63.9 | 52.4 | 76.8 | 58.3 |
| CDAN | 50.7 | 70.6 | 76.0 | 57.6 | 70.0 | 70.0 | 57.4 | 50.9 | 77.3 | 70.9 | 56.7 | 81.6 | 65.8 |
| MDD | 54.9 | 73.7 | 77.8 | 60.0 | 71.4 | 71.8 | 61.2 | 53.6 | 78.1 | 72.5 | 60.2 | 82.3 | 68.1 |
| $f$-DAL | 54.7 | 71.7 | 77.8 | 61.0 | 72.6 | 72.2 | 60.8 | 53.4 | 80.0 | 73.3 | 60.6 | 83.8 | 68.5 |
| $f$-DD ($\chi^2$-DD) | 55.2 | 68.9 | 79.0 | 62.3 | 73.7 | 73.4 | 62.5 | 53.6 | $\mathbf{81.3}$ | 74.8 | 61.0 | 84.1 | 69.2 |
| Ours (Affine) | $\mathbf{55.7}$ | $\mathbf{74.2}$ | $\mathbf{79.1}$ | $\mathbf{62.8}$ | $\mathbf{73.8}$ | $\mathbf{73.8}$ | $\mathbf{63.1}$ | $\mathbf{54.0}$ | 81.1 | $\mathbf{75.2}$ | $\mathbf{61.5}$ | $\mathbf{84.3}$ | $\mathbf{69.9}$ |

*Table 3.* Target accuracy (%). Left block: **Digits**; right block: **VisDA-2017 (ResNet-101)**.

| Method | Digits | | | VisDA-2017 |
|---|---|---|---|---|
| | M$\rightarrow$U | U$\rightarrow$M | Avg | Avg |
| DANN | 91.8 | 94.7 | 93.3 | 57.4 |
| $f$-DAL ($\chi^2$) | 95.3 | 97.3 | 96.3 | 74.3 |
| $f$-DD ($\chi^2$) | 95.4 | 97.3 | 96.4 | 76.2 |
| **Ours (Affine)** | **95.8** | **97.9** | **96.9** | **78.9** |

*Table 4.* Comparison on three benchmarks

| Method | Office-31 | Office-Home | Digits |
|---|---|---|---|
| $f$-DAL | 89.5 | 68.5 | 96.3 |
| $f$-DD | 89.9 | 69.2 | 96.4 |
| Power | 89.8 | 69.4 | 96.5 |
| Affine | **90.4** | **69.9** | **96.9** |
| Exponential | 89.8 | 69.7 | 96.7 |
| Sigmoid | 89.9 | 69.5 | 96.8 |

Figure 1(a) plots accuracy versus $\beta$: performance rises sharply from Hellinger–like values ($\beta \approx 0.5$) to the Pearson point ($\beta = 2$), then slowly decays beyond. Figure 1(b) shows the evolution of the learned $\beta$ over training: both transfer directions converge to distinct optima, indicating task-specific divergence tuning. $\beta$ stabilizes after roughly ten epochs, so we confine its optimization to the first ten epochs to reduce computational overhead and prevent potential overfitting. Table 5 shows adaptive–$\beta$ achieves $96.1\% \pm 0.1\%$ (M$\rightarrow$U), $98.1\% \pm 0.1\%$ (U$\rightarrow$M), and $97.1\% \pm 0.1\%$ avg, versus $96.9\% \pm 0.1\%$ for the fixed Pearson $\chi^2$ point and $97.1\% \pm 0.1\%$ for the freeze variant. These gains indicate that data–driven divergence selection improves over a single pre-selected $\beta$. Additional results on the more challenging Office-Home dataset, along with adaptive strategies, are provided in Appendix I.6.

**Feature Representation Quality.** Qualitative comparison of the learned embeddings (for MNIST$\rightarrow$USPS) under DANN, $f$-DAL, sigmoid-$\tau$, and affine-$\tau$ is provided in Appendix I.7.

## 6. Other Related Work

**Distributional Divergence Measures** Distributional divergences quantify how one probability law differs from another (Amari, 2009). Classic examples include the KL divergence—pervasive in information theory, statistics, and machine learning (Kullback & Leibler, 1951)—and its symmetric Jensen–Shannon variant (Nielsen, 2019). Total variation distance measures the maximum pointwise discrepancy (Devroye et al., 2018), while the Wasserstein distance enjoys favorable mathematical properties (Panaretos & Zemel, 2019). These metrics underpin tasks in statistical inference (Pardo, 2006), deep learning (Cilingir et al., 2020), and data

*Table 5.* Adaptive Cressie–Read $\beta$ Selection on Digits (mean $\pm$ std over 3 runs).

| Method | M$\to$U | U$\to$M | Avg |
|---|---|---|---|
| Fixed $\beta = 2.0$ (Pearson $\chi^2$) | $95.8 \pm 0.2$ | $97.9 \pm 0.1$ | $96.9 \pm 0.1$ |
| Fully Adaptive-$\beta$ | $\mathbf{96.1} \pm 0.1$ | $98.1 \pm 0.1$ | $\mathbf{97.1} \pm 0.1$ |
| Adaptive-$\beta$ (freeze after 10ep) | $95.9 \pm 0.2$ | $\mathbf{98.2} \pm 0.2$ | $\mathbf{97.1} \pm 0.1$ |

mining (Kashyap et al., 2021). More recently, generalizations such as the $\alpha$-divergence and the broad family of $f$-divergences have been introduced, with advances in efficient high-dimensional estimation (Moon & Hero, 2014; Rubenstein et al., 2019) and applications spanning information geometry (Amari & Cichocki, 2010), information theory (Sason & Verdú, 2016), machine learning (Yu et al., 2020), and statistics (Nguyen et al., 2009; Zhang et al., 2022). In this work, we exploit a Tighter-VR of $f$-divergence to drive our UDA framework.

**Divergence-Guided Domain Adaptation** Domain adaptation often employs distributional divergences—such as maximum mean discrepancy (Yan et al., 2017), KL divergence (Nguyen et al., 2022), Cauchy-Schwarz divergence (Yin et al., 2024)—to align source and target feature representations. Adversarial approaches (Ganin et al., 2016) further reduce domain shift by training a domain classifier to minimize divergence during learning (Acuna et al., 2021). In particular, $f$-divergence–guided methods extend this idea, directly minimizing an $f$-divergence criterion to improve tasks ranging from image classification (Wang & Mao, 2024) to sentiment analysis (Cheng et al., 2021). Domain shift also arises in reliability settings such as RUL prediction (Zhu et al., 2026). However, these approaches typically fix the divergence a priori, lacking the flexibility to adapt to varying dataset characteristics. In contrast, our framework dynamically tunes the divergence measure in response to data heterogeneity.

## 7. Conclusion

We proposed a UDA framework that tightens variational $f$-divergence estimation with a learnable transform and adaptively selects the divergence family from data, addressing the fact that a single fixed divergence can be suboptimal under heterogeneous shifts. The resulting discrepancy provides a sharper alignment signal while recovering prior fixed-divergence methods as special cases. We further derived a target-risk bound with finite-sample guarantees, linking transformed source risk, the tighter discrepancy term, and an ideal-hypothesis residual. Experiments on four benchmarks show consistent improvements over strong baselines, and ablations confirm that both the tighter representation and the learned transform contribute to the gains.

**Limitations.** Our guarantees rely on regularity assumptions on $\tau$ and can be conservative with highly expressive critics. Although we limit overhead by learning the divergence parameter only in early epochs and then freezing it, stability may still depend on the schedule and hyperparameters. As with other variational discrepancy estimators, performance can degrade under critic mis-specification or high-variance gradients in high dimensions. Our evaluation is restricted to closed-set image classification; extending to larger-scale and other adaptation settings remains future work.

## Acknowledgements

This work was supported in part by the Sichuan Science and Technology Program under Grant 2024ZYD0135, in part by the National Natural Science Foundation of China under Grants 12071372 and 12201395, and in part by the Fundamental Research Funds for the Central Universities (JBK2507005).

## Impact Statement

This work aims to improve the reliability of unsupervised domain adaptation by learning a more informative cross-domain discrepancy and selecting an appropriate divergence for heterogeneous shifts. The primary expected benefit is better robustness when deploying models across changing data environments, which can reduce performance drops and the need for costly target-domain labels. Potential negative impacts include misuse for large-scale profiling or surveillance if applied to sensitive imagery, and unintended bias amplification if the target domain encodes unfair correlations; our method does not by itself detect or remove such biases. The approach adds some computational overhead during training (though it can be curtailed by early stopping/freezing of divergence selection), which may increase energy use for large models. We encourage practitioners to evaluate subgroup performance, monitor distribution shift in deployment, and apply this method only in settings consistent with applicable privacy, fairness, and data-governance requirements.

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

# Appendix Contents

# A. Summary of Notations

*Table 6.* Summary of notations

| Notation | Definition |
| --- | --- |
| $\mathcal{X}, \mathcal{Y}$ | Input space and output space |
| $\ell : \mathcal{Y} \times \mathcal{Y} \to \mathbb{R}^+$ | Loss function |
| $f_s, f_t$ | Labeling functions for the source and target domains |
| $\{(x_i^s, y_i^s)\}_{i=1}^{n_s}$ | Source observations with sample size $n_s$ |
| $\{x_i^t\}_{i=1}^{n_t}$ | Target observations with sample size $n_t$ |
| $\mathbb{P}_s, \mathbb{P}_t$ | Probability distributions of the source and target domains |
| $h, \mathcal{H}$ | Hypothesis and hypothesis space |
| $R_{\mathbb{P}_s}^{\ell}(h)$ | $\mathbb{E}_{\mathbb{P}_s}[\ell(h(x), f_s(x))]$: Risk function for the source domain |
| $R_{\mathbb{P}_t}^{\ell}(h)$ | $\mathbb{E}_{\mathbb{P}_t}[\ell(h(x), f_t(x))]$: Risk function for the target domain |
| $\mathcal{G}_s$ | Labeled source loss class $\{x \mapsto \ell(h(x), f_s(x)) : h \in \mathcal{H}\}$ |
| $\mathcal{G}_\Delta$ | Pairwise disagreement loss class $\{x \mapsto \ell(h(x), h'(x)) : h, h' \in \mathcal{H}\}$ |
| $\mathcal{L}_\mathcal{H}$ | Enlarged loss class $\mathcal{G}_s \cup \mathcal{G}_\Delta$ |
| $\mathcal{R}(\mathcal{L}_\mathcal{H}, S)$ | Rademacher complexity of $\mathcal{L}_\mathcal{H}$ with respect to sample $S$ |
| $\mathcal{C}(\mathcal{L}_\mathcal{H}, \epsilon, \rho)$ | $\epsilon$-covering number of $\mathcal{L}_\mathcal{H}$ with respect to metric $\rho$ |

# B. Comparison with $f$-DAL, $f$-DD, and Ours

**Summary.** Table 7 summarizes the key design choices and theoretical properties of $f$-DAL (classical Fenchel variational bound), $f$-DD (fixed scaling transform), and Ours (Tighter-VR with a learnable transform and adaptive divergence selection). The underlying $f$-divergence variational form is recalled in Eq. (2), and its tighter representation with a transform $\tau$ is given in Eq. (3); the UDA specialization via the discrepancy $D_{f,\tau}^{h,\mathcal{H}}$ is given in Eq. (4). Ours further couples it with likelihood-driven, data-adaptive selection over the Cressie–Read $\beta$ family, with the $\alpha$-family handled analogously (Sec. 4, Eqs. (11)–(12)). $f$-DD appears as the special case $\tau(x) = ax$ (positive, fixed scale); $f$-DAL corresponds to the identity $\tau(x) = x$.[1]

## B.1. Qualitative Comparison with Closest Prior Methods

$f$-DAL uses the classical Fenchel variational representation and yields the baseline lower bound on $D_f$. $f$-DD tightens this by fixing a scale transform inside the variational form, $\tau(x) = ax$, which (for suitable $a > 0$) produces a tighter bound than the identity case—yet remains task-agnostic and fixed across datasets. In contrast, Ours uses the Tighter-VR family by learning $\tau$ (restricted to a stable, monotone, $L_1$-Lipschitz class) jointly with the encoder/classifier, and augments it with a data-driven selection over the Cressie–Read $\beta$ family via a likelihood criterion, with the $\alpha$-family available as an analogous parameterized choice. This generalizes the prior two: $\tau = x$ recovers $f$-DAL; $\tau(x) = ax$ recovers $f$-DD; learning $\tau$ explores a richer transform family tuned to the observed shift. Empirically, this yields consistent (though modest) gains on Office-Home and a larger margin on VisDA-2017.

## B.2. Why Our Discrepancy is Tighter

**Setup.** Given convex $f$ with $f(1) = 0$, the classical variational form is

$$D_f(P \| Q) = \sup_{g \in \mathcal{M}(\mathcal{X})} \left\{ \mathbb{E}_P[g(X)] - \mathbb{E}_Q[f^*(g(X))] \right\},$$

while the *tighter variational representation* introduces a scalar transform $\tau$ from an admissible family $\mathcal{T}$:

$$D_f(P \| Q) = \sup_g \sup_{\tau \in \mathcal{T}} \left\{ \mathbb{E}_P[\tau \circ g(X)] - \mathbb{E}_Q[f^*(\tau \circ g(X))] \right\}.$$

---

[1]See Eq. (2) for the classical variational form; Eq. (3) for Tighter-VR and $D_{f,\tau} \le D_f$; Remark 3.1 shows that restricting $\tau$ to translations or positive scalings recovers prior discrepancies; Eq. (4) defines $D_{f,\tau}^{h,\mathcal{H}}$ for UDA; Sec. 4 and Eqs. (11)–(12) detail the adaptive Cressie–Read $\beta$ selection mechanism, while the $\alpha$-family can be handled by the same principle.

*Table 7.* Brief comparison of $f$-**DAL**, $f$-**DD**, and **Ours**. Details are in Appx. B.

| Aspect | $f$-**DAL** | $f$-**DD** | **Ours** |
|---|---|---|---|
| VR form | Fenchel VR | Tight-VR | Tighter-VR |
| Tight vs $D_f$ | baseline | fixed tightening | learnable tighter |
| Transform $\tau$ | identity | fixed scale | monotone, $L_1$-Lipschitz, learned |
| Learnable $\tau$ | ✗ | ✗ | ✓ |
| Adaptive divergence | ✗ | ✗(fixed) | ✓(Adaptive) |
| Covers prior | — | — | recovers $f$-DAL / $f$-DD |

For each fixed $\tau$, define the lower bound

$$D_{f,\tau}(P\|Q) := \sup_g \left\{ \mathbb{E}_P[\tau \circ g(X)] - \mathbb{E}_Q\left[f^*(\tau \circ g(X))\right] \right\}.$$

For any $\tau \in \mathcal{T}$, $D_{f,\tau}(P\|Q) \le D_f(P\|Q)$. The choice $\tau_{\mathrm{id}}(x) = x$ recovers the classical $f$-DAL bound, while $\tau_a(x) = ax$ with $a > 0$ corresponds to the fixed-scale $f$-DD family. Since $\tau_{\mathrm{id}} = \tau_{a=1}$, optimizing over positive scales contains the identity case; optimizing over a richer admissible family contains both fixed-scale and identity cases, while still being upper bounded by $D_f$:

$$D_{f,\tau_{\mathrm{id}}}(P\|Q) \ \le\ \sup_{a>0} D_{f,\tau_a}(P\|Q) \ \le\ \sup_{\tau \in \mathcal{T}} D_{f,\tau}(P\|Q) \ \le\ D_f(P\|Q).$$

For any fixed scale $a > 0$, the guaranteed relation is

$$D_{f,\tau_a}(P\|Q) \ \le\ \sup_{\tau \in \mathcal{T}} D_{f,\tau}(P\|Q) \ \le\ D_f(P\|Q).$$

**Simple proof sketch of the tightness chain.** Let $\phi(g, \tau) := \mathbb{E}_P[\tau \circ g] - \mathbb{E}_Q[f^*(\tau \circ g)]$. Then $D_f = \sup_\tau \sup_g \phi(g, \tau)$ and $D_{f,\tau} = \sup_g \phi(g, \tau)$. Since $\tau_{\mathrm{id}}(x) = x$ is the special case $\tau_{a=1}(x) = ax$ within the fixed-scale family $\{\tau_a : a > 0\}$, we have

$$\sup_g \phi(g, \tau_{\mathrm{id}}) \le \sup_{a>0} \sup_g \phi(g, \tau_a).$$

If the admissible family $\mathcal{T}$ contains these positive scalings, then

$$\sup_{a>0} \sup_g \phi(g, \tau_a) \le \sup_{\tau \in \mathcal{T}} \sup_g \phi(g, \tau) \le D_f.$$

For any fixed $a > 0$, one only has $D_{f,\tau_a} \le \sup_{\tau \in \mathcal{T}} D_{f,\tau} \le D_f$. In our algorithm, $\tau$ is learned over a parametric subset of $\mathcal{T}$ that contains the identity and fixed-scale maps, so the optimized transform family can match or improve over $f$-DAL and fixed-scale $f$-DD instances. For UDA, restricting $\tau$ to translations/positive scalings recovers prior discrepancies (Remark 3.1); learning over a richer $\tau$ family therefore tightens the discrepancy attainable in Eq. (4).

**Practical evidence.** On VisDA-2017, our affine-$\tau$ estimator reaches 78.9%, a +2.7pp improvement over the strongest $f$-DD baseline; on Office-Home, adaptive Cressie–Read $\beta$ selection delivers consistent gains. These results align with the theory: a learnable $\tau$ and adaptive $f$ both tighten the effective discrepancy signal used in training.

## C. Relationship Between Generalized $f$-Divergences and the $\beta/\alpha$ Parameterized Families

Let $P$ and $Q$ be probability distributions on $\mathcal{X}$ with densities $p$ and $q$. The classical Csiszár–Morimoto $f$-divergence is defined for a convex generator $f : (0, \infty) \to \mathbb{R}$ with $f(1) = 0$ by

$$\mathcal{D}_f(P\|Q) \ = \ \int_{\mathcal{X}} q(x)\, f\!\left(\tfrac{p(x)}{q(x)}\right) \mathrm{d}x.$$

Throughout this paper, the adaptive $\beta$-family refers to the Cressie–Read power-divergence family, which is itself a Csiszár $f$-divergence family. For $\beta \ne 0, 1$, it arises from the generator (Cressie & Read, 1984)

$$f_\beta(t) \ = \ \frac{t^\beta - \beta t + (\beta - 1)}{\beta(\beta - 1)},$$

leading to

$$D_\beta(P\|Q) = \int_\mathcal{X} q(x) f_\beta\Big(\frac{p(x)}{q(x)}\Big) \mathrm{d}x = \frac{1}{\beta(\beta-1)} \int_\mathcal{X} \Big[p(x)^\beta q(x)^{1-\beta} - \beta\, p(x) + (\beta-1)\, q(x)\Big] \mathrm{d}x,$$

which is the power-divergence form used for the adaptive $f$-divergence selector.

By taking continuous limits of $\beta$, one recovers well-known divergences (up to inessential constants). In particular,

$$\lim_{\beta \to 1} D_\beta(P\|Q) = \int_\mathcal{X} p(x) \log\Big(\tfrac{p(x)}{q(x)}\Big) \mathrm{d}x = D_{\mathrm{KL}}(P\|Q),$$

$$D_2(P\|Q) = \tfrac{1}{2} \int_\mathcal{X} \frac{\big(p(x) - q(x)\big)^2}{q(x)} \mathrm{d}x = \tfrac{1}{2}\, D_{\chi^2}(P\|Q),$$

$$\lim_{\beta \to 0} D_\beta(P\|Q) = \int_\mathcal{X} q(x) \Big[\tfrac{p(x)}{q(x)} - \log\Big(\tfrac{p(x)}{q(x)}\Big) - 1\Big] \mathrm{d}x,$$

which is the Itakura–Saito–type divergence written as an expectation under $Q$ of $t - \log t - 1$ with $t = p/q$. These identities clarify that KL, Pearson $\chi^2$, and Itakura–Saito-type limits appear within the same Csiszár $f$-divergence family indexed by $\beta$.

**Relation to the $\alpha$-divergence family. (Cichocki & Amari, 2010)** The $\alpha$-divergences (including, e.g., Hellinger and KL as special cases) constitute another parameterized $f$-divergence family. While portions of the $\alpha$- and Cressie–Read $\beta$-families can be related through non-linear reparameterizations and escort-measure links (Cichocki & Amari, 2010), there is *no global one-line bijection* that pointwise identifies all members across the entire parameter ranges. Accordingly, we regard the two as complementary parameterizations within the broader class of generalized $f$-divergences; connections are established via appropriate reweightings or limits when required, rather than through a universal algebraic identity.

**Implication for adaptive discrepancy selection.** Our objective is $f$-divergence–agnostic: the same variational pipeline applies to a broad roster of choices, and the particular discrepancy is selected by estimating its parameter(s) from data rather than fixing it a priori. In this view, the Cressie–Read $\beta$-family serves as a tunable $f$-divergence option whose parameter controls sensitivity, while established reparameterization and escort-measure links provide a practical bridge to the $\alpha$-divergence family. Consequently, standard $f$-divergences used in practice, such as Itakura–Saito-type limits, KL, Pearson $\chi^2$, Hellinger, and Jensen–Shannon, are covered by the framework without changing the training recipe.

### C.1. Parameterized $f$-Divergence Families

The framework developed here applies to *any* differentiable, parameterized $f$-divergence family $\{D_{f_\theta} : \theta \in \Theta\}$ whose generator $f_\theta$ admits a Fenchel–Legendre dual $f_\theta^*$.

Consequently, one can embed diverse parametric divergences and estimate their parameters directly from data by likelihood (or asymptotic MSE) criteria, without altering the variational training pipeline. Concrete instances include: (i) the Cressie–Read power-divergence family (Cressie & Read, 1984), whose parameter $\beta$ controls the selected $f$-divergence; (ii) the $\gamma$-divergence (Fujisawa & Eguchi, 2008), for which $\theta = \gamma$ admits a parallel likelihood/asymptotic-MSE selection; and (iii) generalized two-parameter families (e.g., $\alpha$–$\beta$ forms) (Cichocki et al., 2011) where $\theta = (\alpha, \beta)$ can be learned jointly, as is common in robust matrix factorization and related estimation problems. In all cases, the optimization of $\theta$ is decoupled from the choice of representation learner: one may either (a) treat $\theta$ as a hyperparameter selected by

$$\hat{\theta} \in \arg\max_{\theta \in \Theta} \sum_{i=1}^n \log p_\theta(x_i) \quad \text{or} \quad \hat{\theta} \in \arg\min_{\theta \in \Theta}\, D_{f_\theta}(P\|Q),$$

or (b) *jointly* update $\theta$ with model parameters under the same variational objective by differentiating through $f_\theta$ and its dual. Thus, while our empirical illustrations instantiate the Cressie–Read $\beta$-family and discuss the $\alpha$-family as an analogous option, the theoretical guarantees (via the DV form with the logarithmic term and its tighter relaxations) and the training recipe are divergence-agnostic: replacing $f_\theta$ simply swaps the discrepancy while preserving the overall learning and selection mechanism.

# D. Technical Lemmas

**Lemma D.1** (Theorem 3.3 in (Mohri et al., 2012))**.** *Let $\mathcal{G}$ be a family of functions mapping from $\mathcal{Z}$ to $[0, 1]$. Then, for any $\delta > 0$, with probability at least $1 - \delta$ over the draw of an i.i.d. sample $S$ of size $m$, each of the following holds for all $g \in \mathcal{G}$ :*

$$\mathbb{E}[g(z)] \leq \frac{1}{m} \sum_{i=1}^{m} g(z_i) + 2\mathcal{R}_m(\mathcal{G}, \mathcal{Z}) + \sqrt{\frac{\log \frac{1}{\delta}}{2m}}$$

**Lemma D.2** (Lemma 3.4 in (Mohri et al., 2012))**.** *Let $\mathcal{H}$ be a family of functions taking values in $\{-1, +1\}$ and let $\mathcal{G}$ be the family of loss functions associated to $\mathcal{H}$ for the zero-one loss: $\mathcal{G} = \{(x, y) \mapsto 1_{h(x) \neq y} : h \in \mathcal{H}\}$. For any sample $S = ((x_1, y_1), \ldots, (x_m, y_m))$ of elements in $X \times \{-1, +1\}$, let $S_x$ denote its projection over $X : S_x = (x_1, \ldots, x_m)$. Then, the following relation holds between the empirical Rademacher complexities of $\mathcal{G}$ and $\mathcal{H}$ :*

$$\widehat{\mathcal{R}}_S(\mathcal{G}) = \frac{1}{2}\widehat{\mathcal{R}}_{S_x}(\mathcal{H})$$

**Lemma D.3** (Talagrand's contraction lemma, Theorem 26.9 in (Shalev-Shwartz & Ben-David, 2014))**.** *Let $g$ be an $L$-Lipschitz continuous function, and $\mathcal{H}$ is a function class. Then,*

$$\mathcal{R}_n(g \circ \mathcal{H}) \leq L \cdot \mathcal{R}_n(\mathcal{H})$$

**Lemma D.4** (Massart's Lemma (Massart, 2000))**.** *Let $\mathcal{F}$ be a finite class of real-valued functions and let*

$$S = \{z_1, \ldots, z_m\}$$

*be an i.i.d. sample drawn from some distribution. Define*

$$B = \max_{f \in \mathcal{F}} \left(\sum_{i=1}^{m} f(z_i)^2\right)^{1/2}.$$

*Then the empirical Rademacher complexity of $\mathcal{F}$ on $S$ satisfies*

$$\widehat{\mathcal{R}}_m(\mathcal{F}, S) \leq \frac{B\sqrt{2\ln|\mathcal{F}|}}{m}.$$

**Lemma D.5** (Covering Number Bound (Bartlett & Mendelson, 2002))**.** *Let $\mathcal{F}$ be a class of real-valued functions, $S = \{z_1, \cdots, z_m\}$ be a random i.i.d. sample, and $C(\mathcal{F}, \epsilon, \|\cdot\|_{1,\mathcal{S}})$ be the size of the smallest $\epsilon$-cover of $F$ w.r.t. $\|\cdot\|_{1,\mathcal{S}}$, i.e., the covering number. Assuming*

$$\sup_{f \in \mathcal{F}} \left(\frac{1}{m}\sum_{i=1}^{m} f^2(z_i)\right)^{\frac{1}{2}} \leq c$$

*then we have*

$$\hat{R}_m(\mathcal{F}, S) \leq \inf_{\epsilon > 0}\left(\epsilon + \frac{c\sqrt{2}}{\sqrt{m}}\sqrt{\ln C(\mathcal{F}, \epsilon, \|\cdot\|_{1,\mathcal{S}})}\right)$$

**Lemma D.6** (Dudley's Entropy Integral Bound (Shalev-Shwartz & Ben-David, 2014))**.** *Let $\mathcal{F}$ be a class of real-valued functions, $S = \{z_1, \cdots, z_m\}$ be a random i.i.d. sample, and $C(\mathcal{F}, \epsilon, \|\cdot\|_{2,\mathcal{S}})$ be the size of the smallest $\epsilon$-cover of $F$ w.r.t. $\|\cdot\|_{2,\mathcal{S}}$. Assuming*

$$\sup_{f \in \mathcal{F}} \left(\frac{1}{m}\sum_{i=1}^{m} f^2(z_i)\right)^{\frac{1}{2}} \leq c$$

*then we have*

$$\hat{R}_m(\mathcal{F}, S) \leq \inf_{\epsilon \in [0, c/2]}\left(4\epsilon + \frac{12}{\sqrt{m}}\int_{\epsilon}^{c/2}\sqrt{\ln C(\mathcal{F}, \nu, \|\cdot\|_{2,\mathcal{S}})}\mathrm{d}\nu\right)$$

**Lemma D.7** (Massart's Lemma, Theorem 3.3 in (Mohri et al., 2012)). *Let $\mathcal{H}$ be a binary class of functions, $S = \{z_1, \cdots, z_m\}$ be a random i.i.d. sample, $\Pi_{\mathcal{H}}(m)$ be growth functions of $\mathcal{H}$ at $m$. Then,*

$$\hat{R}_m(\mathcal{H}, S) \leq \sqrt{\frac{2 \ln \Pi_{\mathcal{H}}(m)}{m}}$$

# E. Proofs

We provide complete proofs for the main theorems and supporting lemmas stated in the paper.

## E.1. Proof of Theorem 3.2

**Theorem 3.2.** *Let $h^* = \arg\min_{h \in \mathcal{H}} \{R_s^\ell(h) + R_t^\ell(h)\}$ be the ideal hypothesis for the ordinary, untransformed risks. Assume the transformation $\tau$ is non-decreasing and $L_1$–Lipschitz continuous. Then for any $h \in \mathcal{H}$,*

$$R_t^\ell(h) \leq \tilde{R}_s^\ell(h) + D_{f,\tau}^{h,\mathcal{H}}(\mathbb{P}_t || \mathbb{P}_s) + L_1 R^\ell(h^*),$$

*where*

$$\tilde{R}_s^\ell(h) := \sup_{h' \in \mathcal{H}} \left\{ \mathbb{E}_{\mathbb{P}_s}\big[f^* \circ \tau \circ \ell(h(x), h'(x))\big] + L_1 \mathbb{E}_{\mathbb{P}_s}\big[\ell(h'(x), f_s(x))\big] \right\},$$

*and*

$$R^\ell(h^*) := \mathbb{E}_{\mathbb{P}_s}\big[\ell(h^*(x), f_s(x))\big] + \mathbb{E}_{\mathbb{P}_t}\big[\ell(h^*(x), f_t(x))\big].$$

*Proof of Theorem 3.2.* By the triangle inequality of the loss and the monotonicity of $\tau$, we have

$$\begin{aligned}
R_{\mathbb{P}_t}^\ell(h) - R_{\mathbb{P}_s}^\ell(h) &= \mathbb{E}_{\mathbb{P}_t}\big[\tau \circ \ell(h(x), f_t(x))\big] - \mathbb{E}_{\mathbb{P}_s}\big[\tau \circ \ell(h(x), f_s(x))\big] \\
&\leq \mathbb{E}_{\mathbb{P}_t}\big[\tau\big(\ell(h(x), h^*(x)) + \ell(h^*(x), f_t(x))\big)\big] - \mathbb{E}_{\mathbb{P}_s}\big[\tau \circ \ell(h(x), f_s(x))\big] \\
&\leq \mathbb{E}_{\mathbb{P}_t}\big[\tau \circ \ell(h(x), h^*(x))\big] + L_1 \mathbb{E}_{\mathbb{P}_t}\big[\ell(h^*(x), f_t(x))\big] - \mathbb{E}_{\mathbb{P}_s}\big[\tau \circ \ell(h(x), f_s(x))\big], \quad (14)
\end{aligned}$$

where the last inequality uses the $L_1$-Lipschitz property:

$$\tau(a + b) \leq \tau(a) + L_1 |b|.$$

Next, let $a = \ell(h(x), f_s(x))$, $b = \ell(h(x), h^*(x))$, and $c = \ell(h^*(x), f_s(x))$. Since $\ell$ is a metric, the reverse triangle inequality gives $|a - b| \leq c$. Since $\tau$ is $L_1$-Lipschitz, $|\tau(a) - \tau(b)| \leq L_1|a - b| \leq L_1 c$, hence

$$-\tau \circ \ell(h(x), f_s(x)) \leq -\tau \circ \ell(h(x), h^*(x)) + L_1 \ell(h^*(x), f_s(x)).$$

Taking expectation under $\mathbb{P}_s$ and combining with (14) gives

$$\begin{aligned}
R_{\mathbb{P}_t}^\ell(h) - R_{\mathbb{P}_s}^\ell(h) &\leq \mathbb{E}_{\mathbb{P}_t}\big[\tau \circ \ell(h(x), h^*(x))\big] - \mathbb{E}_{\mathbb{P}_s}\big[\tau \circ \ell(h(x), h^*(x))\big] \\
&\quad + L_1 \mathbb{E}_{\mathbb{P}_t}\big[\ell(h^*(x), f_t(x))\big] + L_1 \mathbb{E}_{\mathbb{P}_s}\big[\ell(h^*(x), f_s(x))\big] \\
&= \mathcal{E}_{st}(h, h^*) + L_1 R^\ell(h^*), \quad (15)
\end{aligned}$$

where

$$\mathcal{E}_{st}(h, h^*) = \mathbb{E}_{\mathbb{P}_t}\big[\tau \circ \ell(h(x), h^*(x))\big] - \mathbb{E}_{\mathbb{P}_s}\big[\tau \circ \ell(h(x), h^*(x))\big].$$

Let $g(x) = f(x + 1)$. Then its Fenchel conjugate satisfies

$$g^*(x) = f^*(x) - x. \quad (16)$$

Moreover, since $f(1) = 0$, we have $f^*(x) \geq x$. Define

$$K_s^\ell(h, h') = \mathbb{E}_{\mathbb{P}_s}[g^* \circ \tau \circ \ell(h(x), h'(x))], \qquad K_s^\ell(h) = \sup_{h' \in \mathcal{H}} \mathbb{E}_{\mathbb{P}_s}[g^* \circ \tau \circ \ell(h(x), h'(x))].$$

We first show that
$$\mathcal{E}_{st}(h, h^*) \leq \mathcal{D}_{f,\tau}^{h,\mathcal{H}} \{\mathbb{P}_t || \mathbb{P}_s\} + K_s^\ell(h).$$

By (16),
$$K_s^\ell(h, h') = \mathbb{E}_{\mathbb{P}_s}[f^* \circ \tau \circ \ell(h(x), h'(x))] - \mathbb{E}_{\mathbb{P}_s}[\tau \circ \ell(h(x), h'(x))].$$

Since $K_s^\ell(h, h') \leq K_s^\ell(h)$, we have
$$
\begin{aligned}
&\mathbb{E}_{\mathbb{P}_t}[\tau \circ \ell(h(x), h'(x))] - \mathbb{E}_{\mathbb{P}_s}[\tau \circ \ell(h(x), h'(x))] - K_s^\ell(h) \\
&\leq \mathbb{E}_{\mathbb{P}_t}[\tau \circ \ell(h(x), h'(x))] - \mathbb{E}_{\mathbb{P}_s}[f^* \circ \tau \circ \ell(h(x), h'(x))] \\
&\leq \sup_{h' \in \mathcal{H}} \left\{ \mathbb{E}_{\mathbb{P}_t}[\tau \circ \ell(h(x), h'(x))] - \mathbb{E}_{\mathbb{P}_s}[f^* \circ \tau \circ \ell(h(x), h'(x))] \right\} \\
&= \mathcal{D}_{f,\tau}^{h,\mathcal{H}} \{\mathbb{P}_t || \mathbb{P}_s\}.
\end{aligned}
$$

Taking $h' = h^*$ yields
$$\mathcal{E}_{st}(h, h^*) \leq K_s^\ell(h) + \mathcal{D}_{f,\tau}^{h,\mathcal{H}} \{\mathbb{P}_t || \mathbb{P}_s\}. \tag{17}$$

Combining (15) and (17) gives
$$R_{\mathbb{P}_t}^\ell(h) - R_{\mathbb{P}_s}^\ell(h) \leq K_s^\ell(h) + \mathcal{D}_{f,\tau}^{h,\mathcal{H}} \{\mathbb{P}_t || \mathbb{P}_s\} + L_1 R^\ell(h^*).$$

Thus,
$$R_{\mathbb{P}_t}^\ell(h) \leq R_{\mathbb{P}_s}^\ell(h) + K_s^\ell(h) + \mathcal{D}_{f,\tau}^{h,\mathcal{H}} \{\mathbb{P}_t || \mathbb{P}_s\} + L_1 R^\ell(h^*). \tag{18}$$

Finally, we bound $R_{\mathbb{P}_s}^\ell(h) + K_s^\ell(h)$ as follows:
$$
\begin{aligned}
&R_{\mathbb{P}_s}^\ell(h) + K_s^\ell(h) \\
=\ & \mathbb{E}_{\mathbb{P}_s}[\tau \circ \ell(h(x), f_s(x))] + \sup_{h' \in \mathcal{H}} \mathbb{E}_{\mathbb{P}_s}[g^* \circ \tau \circ \ell(h(x), h'(x))] \\
=\ & \sup_{h' \in \mathcal{H}} \left\{ \mathbb{E}_{\mathbb{P}_s}[f^* \circ \tau \circ \ell(h(x), h'(x))] + \mathbb{E}_{\mathbb{P}_s}[\tau \circ \ell(h(x), f_s(x))] - \mathbb{E}_{\mathbb{P}_s}[\tau \circ \ell(h(x), h'(x))] \right\} \\
\leq\ & \sup_{h' \in \mathcal{H}} \left\{ \mathbb{E}_{\mathbb{P}_s}[f^* \circ \tau \circ \ell(h(x), h'(x))] + L_1 \mathbb{E}_{\mathbb{P}_s}[\ell(h'(x), f_s(x))] \right\} \\
=:\ & \tilde{R}_{\mathbb{P}_s}^\ell(h). \tag{19}
\end{aligned}
$$

*Justification of the last inequality.* For any $x$, let $a = \ell(h(x), f_s(x))$, $b = \ell(h(x), h'(x))$, and $c = \ell(h'(x), f_s(x))$. Since $\ell$ is a metric, the reverse triangle inequality gives $|a - b| \leq c$. Since $\tau$ is $L_1$-Lipschitz, $|\tau(a) - \tau(b)| \leq L_1|a - b| \leq L_1 c$, hence $\tau(a) - \tau(b) \leq L_1 c$. Taking expectation over $\mathbb{P}_s$ yields
$$\mathbb{E}_{\mathbb{P}_s}[\tau \circ \ell(h, f_s)] - \mathbb{E}_{\mathbb{P}_s}[\tau \circ \ell(h, h')] \leq L_1 \mathbb{E}_{\mathbb{P}_s}[\ell(h', f_s)].$$

Substituting (19) into (18) completes the proof. $\qquad\square$

### E.2. Proof of Lemma 3.4

**Lemma 3.4.** *Assume that the transformation $\tau$ and the Fenchel conjugate function $f^*$ are Lipschitz continuous with Lipschitz constants $L_1$ and $L_2$, respectively. Then, with probability at least $1 - \delta$, the following inequality holds for all $h \in \mathcal{H}$*
$$
\begin{aligned}
&\tilde{R}_{\mathbb{P}_s}^\ell(h) - \tilde{R}_{\widehat{\mathbb{P}}_s}^\ell(h) \\
&\leq\ 2L_1(1 + L_2)\widehat{\mathcal{R}}_{n_s}(\mathcal{L}_\mathcal{H}, S) + O\sqrt{\frac{\log(1/\delta)}{n_s}}
\end{aligned}
$$

*where $\delta > 0$ is a constant, $\widehat{\mathbb{P}}_s$ denotes the empirical measure of the source sample $S$, $\widehat{\mathcal{R}}_{n_s}(\mathcal{L}_\mathcal{H}, S)$ indicates the empirical Rademacher complexity of the enlarged loss class $\mathcal{L}_\mathcal{H}$ w.r.t. the sample $S$.*

*Proof of Lemma 3.4.* For each $h \in \mathcal{H}$, we have

$$
\begin{aligned}
& \tilde{R}^{\ell}_{\mathbb{P}_s}(h) - \tilde{R}^{\ell}_{\widehat{\mathbb{P}}_s}(h) \\
= \ & \sup_{h' \in \mathcal{H}} \left\{ \mathbb{E}_{\mathbb{P}_s}\big[ f^* \circ \tau \circ \ell(h(x), h'(x)) \big] + L_1 \, \mathbb{E}_{\mathbb{P}_s}\big[ \ell(h'(x), f_s(x)) \big] \right\} \\
& - \sup_{h' \in \mathcal{H}} \left\{ \mathbb{E}_{\widehat{\mathbb{P}}_s}\big[ f^* \circ \tau \circ \ell(h(x), h'(x)) \big] + L_1 \, \mathbb{E}_{\widehat{\mathbb{P}}_s}\big[ \ell(h'(x), f_s(x)) \big] \right\} \\
\leq \ & \sup_{h' \in \mathcal{H}} \left| \mathbb{E}_{\mathbb{P}_s}\big[ f^* \circ \tau \circ \ell(h(x), h'(x)) \big] - \mathbb{E}_{\widehat{\mathbb{P}}_s}\big[ f^* \circ \tau \circ \ell(h(x), h'(x)) \big] \right| \\
& + L_1 \sup_{h' \in \mathcal{H}} \left| \mathbb{E}_{\mathbb{P}_s}\big[ \ell(h'(x), f_s(x)) \big] - \mathbb{E}_{\widehat{\mathbb{P}}_s}\big[ \ell(h'(x), f_s(x)) \big] \right| \\
=: \ & A_1 + A_2.
\end{aligned}
\tag{20}
$$

where

$$
\begin{aligned}
A_1 \ &\overset{\text{def}}{=} \ \sup_{h' \in \mathcal{H}} \left| \mathbb{E}_{\mathbb{P}_s}\big[ f^* \circ \tau \circ \ell(h(x), h'(x)) \big] - \mathbb{E}_{\widehat{\mathbb{P}}_s}\big[ f^* \circ \tau \circ \ell(h(x), h'(x)) \big] \right|, \\
A_2 \ &\overset{\text{def}}{=} \ L_1 \sup_{h' \in \mathcal{H}} \left| \mathbb{E}_{\mathbb{P}_s}\big[ \ell(h'(x), f_s(x)) \big] - \mathbb{E}_{\widehat{\mathbb{P}}_s}\big[ \ell(h'(x), f_s(x)) \big] \right|.
\end{aligned}
$$

For the term $A_2$, the class $\{ x \mapsto \ell(h'(x), f_s(x)) : h' \in \mathcal{H} \}$ is contained in $\mathcal{G}_s \subset \mathcal{L}_{\mathcal{H}}$. Hence, by Lemmas D.1 and D.3, with probability at least $1 - \delta$, we have

$$
A_2 \ \leq \ 2 L_1 \widehat{\mathcal{R}}_{n_s}(\mathcal{L}_{\mathcal{H}}, S) + O\sqrt{\frac{\log(1/\delta)}{n_s}}.
\tag{21}
$$

Next, for fixed $h$, the class $\{ x \mapsto \ell(h(x), h'(x)) : h' \in \mathcal{H} \}$ is contained in $\mathcal{G}_\Delta \subset \mathcal{L}_{\mathcal{H}}$. Note that $\tau$ is $L_1$-Lipschitz and $f^*$ is $L_2$-Lipschitz. Hence the composition $f^* \circ \tau$ is $(L_1 L_2)$-Lipschitz. Applying Lemmas D.1 and D.3 again, with probability at least $1 - \delta$, we obtain

$$
\begin{aligned}
A_1 \ &\leq \ 2 \widehat{\mathcal{R}}_{n_s}\big( (f^* \circ \tau) \circ \mathcal{L}_{\mathcal{H}}, S \big) + O\sqrt{\frac{\log(1/\delta)}{n_s}} \\
&\leq \ 2 L_1 L_2 \widehat{\mathcal{R}}_{n_s}(\mathcal{L}_{\mathcal{H}}, S) + O\sqrt{\frac{\log(1/\delta)}{n_s}}.
\end{aligned}
\tag{22}
$$

Combining (20)–(22) yields that, with probability at least $1 - \delta$,

$$
\begin{aligned}
\tilde{R}^{\ell}_{\mathbb{P}_s}(h) - \tilde{R}^{\ell}_{\widehat{\mathbb{P}}_s}(h) \ &\leq \ 2 L_1 L_2 \widehat{\mathcal{R}}_{n_s}(\mathcal{L}_{\mathcal{H}}, S) + 2 L_1 \widehat{\mathcal{R}}_{n_s}(\mathcal{L}_{\mathcal{H}}, S) + O\sqrt{\frac{\log(1/\delta)}{n_s}} \\
&= \ 2 L_1 (1 + L_2) \widehat{\mathcal{R}}_{n_s}(\mathcal{L}_{\mathcal{H}}, S) + O\sqrt{\frac{\log(1/\delta)}{n_s}}.
\end{aligned}
$$

This completes the proof. $\square$

### E.3. Proof of Lemma 3.5

**Lemma 3.5.** *Assume that the assumptions of Lemma 3.4 hold. Then, with probability at least $1 - \delta$, the following inequality holds*

$$
\begin{aligned}
& \mathcal{D}^{h, \mathcal{H}}_{f, \tau} \{ \mathbb{P}_t \| \mathbb{P}_s \} - \mathcal{D}^{h, \mathcal{H}}_{f, \tau} \left\{ \widehat{\mathbb{P}}_t \| \widehat{\mathbb{P}}_s \right\} \\
\leq \ & 2 L_1 \widehat{\mathcal{R}}_{n_t}(\mathcal{L}_{\mathcal{H}}, T) + 2 L_1 L_2 \widehat{\mathcal{R}}_{n_s}(\mathcal{L}_{\mathcal{H}}, S) \\
& + O\sqrt{\frac{\log(1/\delta)}{n_s}} + O\sqrt{\frac{\log(1/\delta)}{n_t}}
\end{aligned}
$$

where $\widehat{\mathcal{R}}_{n_s}(\mathcal{L}_{\mathcal{H}}, S)$ and $\widehat{\mathcal{R}}_{n_t}(\mathcal{L}_{\mathcal{H}}, T)$ indicate the empirical Rademacher complexity of the enlarged loss class $\mathcal{L}_{\mathcal{H}}$ w.r.t. the samples $S$ and $T$, respectively.

*Proof of Lemma 3.5.* A simple reorganization gives that

$$
\begin{aligned}
& \mathcal{D}_{f,\tau}^{h,\mathcal{H}} \{\mathbb{P}_t \| \mathbb{P}_s\} - \mathcal{D}_{f,\tau}^{h,\mathcal{H}} \left\{ \widehat{\mathbb{P}}_t \| \widehat{\mathbb{P}}_s \right\} \\
= \; & \sup_{h' \in \mathcal{H}} \{ \mathbb{E}_{\mathbb{P}_t}[\tau \circ \ell(h(x), h'(x))] - \mathbb{E}_{\mathbb{P}_s}[f^* \circ \tau \circ \ell(h(x), h'(x))] \} \\
& - \sup_{h' \in \mathcal{H}} \left\{ \mathbb{E}_{\widehat{\mathbb{P}}_t}[\tau \circ \ell(h(x), h'(x))] - \mathbb{E}_{\widehat{\mathbb{P}}_s}[f^* \circ \tau \circ \ell(h(x), h'(x))] \right\} \\
\leq \; & \sup_{h' \in \mathcal{H}} \left\{ \mathbb{E}_{\mathbb{P}_t}[\tau \circ \ell(h(x), h'(x))] - \mathbb{E}_{\widehat{\mathbb{P}}_t}[\tau \circ \ell(h(x), h'(x))] \right\} \\
& - \sup_{h' \in \mathcal{H}} \left\{ \mathbb{E}_{\mathbb{P}_s}[f^* \circ \tau \circ \ell(h(x), h'(x))] - \mathbb{E}_{\widehat{\mathbb{P}}_s}[f^* \circ \tau \circ \ell(h(x), h'(x))] \right\} \\
\leq \; & \sup_{h' \in \mathcal{H}} \left\{ \left| \mathbb{E}_{\mathbb{P}_t}[\tau \circ \ell(h(x), h'(x))] - \mathbb{E}_{\widehat{\mathbb{P}}_t}[\tau \circ \ell(h(x), h'(x))] \right| \right\} \\
& + \sup_{h' \in \mathcal{H}} \left\{ \left| \mathbb{E}_{\mathbb{P}_s}[f^* \circ \tau \circ \ell(h(x), h'(x))] - \mathbb{E}_{\widehat{\mathbb{P}}_s}[f^* \circ \tau \circ \ell(h(x), h'(x))] \right| \right\} \qquad (23)
\end{aligned}
$$

For fixed $h$, the empirical-process classes in Eq. (23) are contained in $\mathcal{G}_\Delta \subset \mathcal{L}_{\mathcal{H}}$. Note that $\tau$ and $f^*$ are Lipschitz with constants $L_1$ and $L_2$, respectively. Then, from Lemmas D.1 and D.3, with probability at least $1 - \delta$, we have

$$
\begin{aligned}
& \sup_{h' \in \mathcal{H}} \left\{ \left| \mathbb{E}_{\mathbb{P}_t}[\tau \circ \ell(h(x), h'(x))] - \mathbb{E}_{\widehat{\mathbb{P}}_t}[\tau \circ \ell(h(x), h'(x))] \right| \right\} \\
\leq \; & 2L_1 \widehat{\mathcal{R}}_{n_t}(\mathcal{L}_{\mathcal{H}}, T) + O\sqrt{\frac{\log(1/\delta)}{n_t}}
\end{aligned} \qquad (24)
$$

and

$$
\begin{aligned}
& \sup_{h' \in \mathcal{H}} \left\{ \left| \mathbb{E}_{\mathbb{P}_s}[f^* \circ \tau \circ \ell(h(x), h'(x))] - \mathbb{E}_{\widehat{\mathbb{P}}_s}[f^* \circ \tau \circ \ell(h(x), h'(x))] \right| \right\} \\
\leq \; & 2L_1 L_2 \widehat{\mathcal{R}}_{n_s}(\mathcal{L}_{\mathcal{H}}, S) + O\sqrt{\frac{\log(1/\delta)}{n_s}}
\end{aligned} \qquad (25)
$$

Substituting Eqs. (24) and (25) into Eq. (23) completes the proof.

$\qquad\qquad\qquad\qquad\qquad\qquad\qquad\qquad\qquad\qquad\qquad\qquad\qquad\qquad\qquad\qquad\qquad\qquad\qquad\qquad\square$

### E.4. Proof of Theorem 3.6

*Proof of Theorem 3.6.* Starting from Theorem 3.2, replace the population transformed source-risk term by its empirical counterpart using Lemma 3.4, and replace the population Tighter-VR discrepancy by its empirical counterpart using Lemma 3.5. Adding the two source-side Rademacher terms gives

$$
2L_1(1 + L_2)\widehat{\mathcal{R}}_{n_s}(\mathcal{L}_{\mathcal{H}}, S) + 2L_1 L_2 \widehat{\mathcal{R}}_{n_s}(\mathcal{L}_{\mathcal{H}}, S) = 2L_1(1 + 2L_2)\widehat{\mathcal{R}}_{n_s}(\mathcal{L}_{\mathcal{H}}, S),
$$

while the target-side discrepancy term contributes $2L_1 \widehat{\mathcal{R}}_{n_t}(\mathcal{L}_{\mathcal{H}}, T)$. Combining the concentration remainders yields the stated $O(\sqrt{\log(1/\delta)/n_s} + \sqrt{\log(1/\delta)/n_t})$ term, and the ideal-risk residual is unchanged. $\qquad\square$

### E.5. Proof of Theorem 3.7

**Theorem 3.7.** *Assume that the enlarged loss class $\mathcal{L}_{\mathcal{H}}$ is finite. Let*

$$
B_s = \max_{g \in \mathcal{L}_{\mathcal{H}}} \left( \sum_{i=1}^{n_s} g(x_i^s)^2 \right)^{1/2}
$$

$$
B_t = \max_{g \in \mathcal{L}_{\mathcal{H}}} \left( \sum_{i=1}^{n_t} g(x_i^t)^2 \right)^{1/2}
$$

*Then, with probability at least $1 - \delta$, we have*

$$R^{\ell}_{\mathbb{P}_t}(h) \leq \tilde{R}^{\ell}_{\widehat{\mathbb{P}}_s}(h) + 2L_1 \left[ (1 + 2L_2) \frac{B_s \sqrt{2 \ln |\mathcal{L}_{\mathcal{H}}|}}{n_s} + \frac{B_t \sqrt{2 \ln |\mathcal{L}_{\mathcal{H}}|}}{n_t} \right]$$

$$+ \mathcal{D}^{h,\mathcal{H}}_{f,\tau} \left\{ \widehat{\mathbb{P}}_t || \widehat{\mathbb{P}}_s \right\} + L_1 R^{\ell}(h^*) + O\left( \sqrt{\frac{\log(1/\delta)}{n_s}} + \sqrt{\frac{\log(1/\delta)}{n_t}} \right) \tag{26}$$

*Proof of Theorem 3.7.* From Lemma D.4, we have

$$\widehat{\mathcal{R}}_{n_s}(\mathcal{L}_{\mathcal{H}}, S) \leq \frac{B_s \sqrt{2 \ln |\mathcal{L}_{\mathcal{H}}|}}{n_s} \tag{27a}$$

$$\widehat{\mathcal{R}}_{n_t}(\mathcal{L}_{\mathcal{H}}, T) \leq \frac{B_t \sqrt{2 \ln |\mathcal{L}_{\mathcal{H}}|}}{n_t} \tag{27b}$$

Substituting Eq. (27) into Eq. (6) completes the proof. □

### E.6. Proof of Theorem 3.10

**Theorem 3.10.** *Assume that the assumptions of Theorem 3.6 hold and there exists a constant $c > 0$ such that $\|g\|_{2,S} \leq c$ and $\|g\|_{2,T} \leq c$ for all $g \in \mathcal{L}_{\mathcal{H}}$. Then, with probability at least $1 - \delta$, we have*

$$R^{\ell}_{\mathbb{P}_t}(h) \leq \tilde{R}^{\ell}_{\widehat{\mathbb{P}}_s}(h) + 2L_1 \left[ (1 + 2L_2) \times \inf_{\varepsilon > 0} \left( \varepsilon + \frac{c\sqrt{2}}{\sqrt{n_s}} \sqrt{\ln \mathcal{C}(\mathcal{L}_{\mathcal{H}}, \varepsilon, \| \cdot \|_{1,S})} \right) \right.$$

$$\left. + \inf_{\varepsilon > 0} \left( \varepsilon + \frac{c\sqrt{2}}{\sqrt{n_t}} \sqrt{\ln \mathcal{C}(\mathcal{L}_{\mathcal{H}}, \varepsilon, \| \cdot \|_{1,T})} \right) \right] + \mathcal{D}^{h,\mathcal{H}}_{f,\tau} \left\{ \widehat{\mathbb{P}}_t || \widehat{\mathbb{P}}_s \right\} + L_1 R^{\ell}(h^*)$$

$$+ O\left( \sqrt{\frac{\log(1/\delta)}{n_s}} + \sqrt{\frac{\log(1/\delta)}{n_t}} \right) \tag{28}$$

*Proof of Theorem 3.10.* Note that

$$\|g\|_{2,S} \leq c, \quad \|g\|_{2,T} \leq c \quad \text{for all } g \in \mathcal{L}_{\mathcal{H}}.$$

Then Lemma D.5 gives

$$\widehat{\mathcal{R}}_{n_s}(\mathcal{L}_{\mathcal{H}}, S) \leq \inf_{\varepsilon > 0} \left( \varepsilon + \frac{c\sqrt{2}}{\sqrt{n_s}} \sqrt{\ln \mathcal{C}(\mathcal{L}_{\mathcal{H}}, \varepsilon, \| \cdot \|_{1,S})} \right) \tag{29a}$$

$$\widehat{\mathcal{R}}_{n_t}(\mathcal{L}_{\mathcal{H}}, T) \leq \inf_{\varepsilon > 0} \left( \varepsilon + \frac{c\sqrt{2}}{\sqrt{n_t}} \sqrt{\ln \mathcal{C}(\mathcal{L}_{\mathcal{H}}, \varepsilon, \| \cdot \|_{1,T})} \right) \tag{29b}$$

The proof follows by substituting Eq. (29) into Eq. (6). □

### E.7. Proof of Theorem 3.11

**Theorem 3.11.** *Assume that the assumptions of Theorem 3.10 hold. Then, with probability at least $1 - \delta$, we have*

$$R^{\ell}_{\mathbb{P}_t}(h) \leq \tilde{R}^{\ell}_{\widehat{\mathbb{P}}_s}(h) + 2L_1 \left[ (1 + 2L_2) \times \inf_{\varepsilon \in [0, c/2]} \left( 4\varepsilon + \frac{12}{\sqrt{n_s}} \int_{\varepsilon}^{c/2} \sqrt{\ln \mathcal{C}(\mathcal{L}_{\mathcal{H}}, \eta, \| \cdot \|_{2,S})} \, d\eta \right) \right.$$

$$\left. + \inf_{\varepsilon \in [0, c/2]} \left( 4\varepsilon + \frac{12}{\sqrt{n_t}} \int_{\varepsilon}^{c/2} \sqrt{\ln \mathcal{C}(\mathcal{L}_{\mathcal{H}}, \eta, \| \cdot \|_{2,T})} \, d\eta \right) \right] + \mathcal{D}^{h,\mathcal{H}}_{f,\tau} \left\{ \widehat{\mathbb{P}}_t || \widehat{\mathbb{P}}_s \right\} + L_1 R^{\ell}(h^*)$$

$$+ O\left( \sqrt{\frac{\log(1/\delta)}{n_s}} + \sqrt{\frac{\log(1/\delta)}{n_t}} \right) \tag{30}$$

*Proof of Theorem 3.11.* From Dudley's entropy integral bound (Shalev-Shwartz & Ben-David, 2014), we have

$$\widehat{\mathcal{R}}_{n_s}(\mathcal{L}_{\mathcal{H}}, S) \leq \inf_{\varepsilon \in [0, c/2]} \left( 4\varepsilon + \frac{12}{\sqrt{n_s}} \int_{\varepsilon}^{c/2} \sqrt{\ln \mathcal{C}(\mathcal{L}_{\mathcal{H}}, \eta, \|\cdot\|_{2,S})} \, d\eta \right) \tag{31a}$$

$$\widehat{\mathcal{R}}_{n_t}(\mathcal{L}_{\mathcal{H}}, T) \leq \inf_{\varepsilon \in [0, c/2]} \left( 4\varepsilon + \frac{12}{\sqrt{n_t}} \int_{\varepsilon}^{c/2} \sqrt{\ln \mathcal{C}(\mathcal{L}_{\mathcal{H}}, \eta, \|\cdot\|_{2,T})} \, d\eta \right) \tag{31b}$$

Combining Eqs. (6) and (31) completes the proof. □

## E.8. Proof of Theorem 3.12

**Theorem 3.12.** *Let $VC(\mathcal{L}_{\mathcal{H}}, T)$ and $VC(\mathcal{L}_{\mathcal{H}}, S)$ be the VC-dimensions of the enlarged loss class $\mathcal{L}_{\mathcal{H}}$ associated with samples $T$ and $S$, respectively. We assume that the conditions of Theorem 3.6 hold, $n_t \geq VC(\mathcal{L}_{\mathcal{H}}, T)$ and $n_s \geq VC(\mathcal{L}_{\mathcal{H}}, S)$. Then, with probability at least $1 - \delta$, we have*

$$R_{\mathbb{P}_t}^{\ell}(h) \leq \tilde{R}_{\widehat{\mathbb{P}}_s}^{\ell}(h) + 2L_1 \left[ (1 + 2L_2) \times \sqrt{\frac{2VC(\mathcal{L}_{\mathcal{H}}, S) \log(en_s/VC(\mathcal{L}_{\mathcal{H}}, S))}{n_s}} \right.$$

$$\left. + \sqrt{\frac{2VC(\mathcal{L}_{\mathcal{H}}, T) \log(en_t/VC(\mathcal{L}_{\mathcal{H}}, T))}{n_t}} \right] + \mathcal{D}_{f,\tau}^{h,\mathcal{H}} \left\{ \widehat{\mathbb{P}}_t \| \widehat{\mathbb{P}}_s \right\} + L_1 \, R^{\ell}(h^*)$$

$$+ O \left( \sqrt{\frac{\log(1/\delta)}{n_s}} + \sqrt{\frac{\log(1/\delta)}{n_t}} \right) \tag{32}$$

*Proof of Theorem 3.12.* Let $\Pi_{\mathcal{L}_{\mathcal{H}}}(n_s)$ and $\Pi_{\mathcal{L}_{\mathcal{H}}}(n_t)$ be growth functions of $\mathcal{L}_{\mathcal{H}}$ evaluated at $n_s$ and $n_t$, respectively. Then, Massart's lemma (Wainwright, 2019) gives that

$$\widehat{\mathcal{R}}_{n_s}(\mathcal{L}_{\mathcal{H}}, S) \leq \sqrt{\frac{2 \ln(\Pi_{\mathcal{L}_{\mathcal{H}}}(n_s))}{n_s}} \tag{33a}$$

$$\widehat{\mathcal{R}}_{n_t}(\mathcal{L}_{\mathcal{H}}, T) \leq \sqrt{\frac{2 \ln(\Pi_{\mathcal{L}_{\mathcal{H}}}(n_t))}{n_t}} \tag{33b}$$

Note that $n_t \geq VC(\mathcal{L}_{\mathcal{H}}, T)$ and $n_s \geq VC(\mathcal{L}_{\mathcal{H}}, S)$. Then the Sauer–Shelah lemma (Mohri et al., 2012) gives

$$\Pi_{\mathcal{L}_{\mathcal{H}}}(n_s) \leq \left( \frac{en_s}{VC(\mathcal{L}_{\mathcal{H}}, S)} \right)^{VC(\mathcal{L}_{\mathcal{H}}, S)} \tag{34a}$$

$$\Pi_{\mathcal{L}_{\mathcal{H}}}(n_t) \leq \left( \frac{en_t}{VC(\mathcal{L}_{\mathcal{H}}, T)} \right)^{VC(\mathcal{L}_{\mathcal{H}}, T)} \tag{34b}$$

Eqs. (33) and (34) lead to

$$\widehat{\mathcal{R}}_{n_s}(\mathcal{L}_{\mathcal{H}}, S) \leq \sqrt{\frac{2VC(\mathcal{L}_{\mathcal{H}}, S) \ln(en_s/VC(\mathcal{L}_{\mathcal{H}}, S))}{n_s}} \tag{35a}$$

$$\widehat{\mathcal{R}}_{n_t}(\mathcal{L}_{\mathcal{H}}, T) \leq \sqrt{\frac{2VC(\mathcal{L}_{\mathcal{H}}, T) \ln(en_t/VC(\mathcal{L}_{\mathcal{H}}, T))}{n_t}} \tag{35b}$$

Substituting Eq. (35) into Eq. (6) completes the proof. □

## E.9. Corollary of Theorem 3.12: Multi-class Extension

**Corollary of Theorem 3.12.** *Let $d_N$ be the Natarajan dimension of $\mathcal{L}_{\mathcal{H}}$ on a $K$-class classification problem. Assume the conditions of Theorem 3.6 hold, and that*

$$n_s \geq d_N, \qquad n_t \geq d_N.$$

*Then with probability at least $1 - \delta$, for any $h \in \mathcal{H}$,*

$$R^\ell_{\mathbb{P}_t}(h) \;\leq\; \tilde{R}^\ell_{\hat{\mathbb{P}}_s}(h)$$

$$+ 2L_1 \left[ (1 + 2L_2) \sqrt{\frac{2\, d_N\, \ln(e\, n_s\, (K-1)/d_N)}{n_s}} \;+\; \sqrt{\frac{2\, d_N\, \ln(e\, n_t\, (K-1)/d_N)}{n_t}} \right]$$

$$+ \mathcal{D}^{h,\mathcal{H}}_{f,\tau}\{\hat{\mathbb{P}}_t \,\|\, \hat{\mathbb{P}}_s\} + L_1\, R^\ell(h^*) + O\Big( \sqrt{\tfrac{\ln(1/\delta)}{n_s}} + \sqrt{\tfrac{\ln(1/\delta)}{n_t}} \Big).$$

*Proof.* Let $\Pi_{\mathcal{L}_\mathcal{H}}(n)$ be the growth function of $\mathcal{L}_\mathcal{H}$ at sample size $n$. By the multiclass Sauer–Shelah lemma (Daniely et al., 2015) (using the Natarajan dimension) we have

$$\Pi_{\mathcal{L}_\mathcal{H}}(n) \;\leq\; \sum_{i=0}^{d_N} \binom{n}{i} (K-1)^i \;\leq\; \left( \frac{e\, n\, (K-1)}{d_N} \right)^{d_N}.$$

Massart's lemma then gives for source and target samples,

$$\widehat{\mathcal{R}}_{n_s}(\mathcal{L}_\mathcal{H}, S) \leq \sqrt{\frac{2\ln\big(\Pi_{\mathcal{L}_\mathcal{H}}(n_s)\big)}{n_s}} \;\leq\; \sqrt{\frac{2\, d_N\, \ln(e\, n_s\, (K-1)/d_N)}{n_s}}, \tag{36a}$$

$$\widehat{\mathcal{R}}_{n_t}(\mathcal{L}_\mathcal{H}, T) \leq \sqrt{\frac{2\ln\big(\Pi_{\mathcal{L}_\mathcal{H}}(n_t)\big)}{n_t}} \;\leq\; \sqrt{\frac{2\, d_N\, \ln(e\, n_t\, (K-1)/d_N)}{n_t}}. \tag{36b}$$

Substituting these bounds into the generalization bound of Theorem 3.6 (cf. Eq. (6)) completes the proof. $\qquad\square$

### E.10. Derivation of the Normalizing Constant $z(x^s, \beta, \phi)$

This subsection derives the normalizer for the Tweedie-linked selection density in Eq. (9). The scalar deviance $\Delta_\beta$ below is used to estimate the parameter $\beta$; after $\beta$ is selected, the discrepancy inserted into the Tighter-VR objective is the Cressie–Read $f$-divergence in Eq. (10). By construction, $z(x^s, \beta, \phi)$ ensures that the selection density integrates to one:

$$z(x^s, \beta, \phi) \;=\; \int_0^\infty \exp\Big\{ k(x^t, \beta) \;-\; \frac{1}{\phi}\Delta_\beta(x^t \| x^s) \Big\}\, \mathrm{d}x^t,$$

where

$$k(x^t, \beta) = \frac{\beta - 1}{2} \ln x^t, \qquad \Delta_\beta(x^t \| x^s) = \frac{1}{\beta(\beta-1)}\Big[ (x^t)^\beta + (\beta-1)(x^s)^\beta - \beta\, x^t\, (x^s)^{\beta-1} \Big].$$

Substituting these into the integrand and gathering terms gives

$$z(x^s, \beta, \phi) = \int_0^\infty (x^t)^{\frac{\beta-1}{2}} \exp\Big\{ -\frac{(x^t)^\beta + (\beta-1)(x^s)^\beta - \beta\, x^t\, (x^s)^{\beta-1}}{\phi\, \beta(\beta-1)} \Big\}\, \mathrm{d}x^t.$$

Pulling out the factor independent of $x^t$, $\exp\{-(x^s)^\beta/(\phi\,\beta)\}$, we obtain

$$z(x^s, \beta, \phi) = \exp\Big\{ -\frac{(x^s)^\beta}{\phi\,\beta} \Big\} \int_0^\infty (x^t)^{\frac{\beta-1}{2}} \exp\Big\{ -\frac{(x^t)^\beta - \beta\, x^t\, (x^s)^{\beta-1}}{\phi\,\beta(\beta-1)} \Big\}\, \mathrm{d}x^t.$$

Now perform the change of variables

$$y = \frac{x^t}{x^s}, \quad x^t = x^s\, y, \quad \mathrm{d}x^t = x^s\, \mathrm{d}y,$$

and note

$$(x^t)^\beta - \beta\, x^t\, (x^s)^{\beta-1} = (x^s)^\beta \big( y^\beta - \beta\, y \big), \quad (x^t)^{\frac{\beta-1}{2}}\, \mathrm{d}x^t = (x^s)^{\frac{\beta+1}{2}}\, y^{\frac{\beta-1}{2}}\, \mathrm{d}y.$$

Hence

$$\boxed{\; z(x^s, \beta, \phi) = \exp\Big\{ -\frac{(x^s)^\beta}{\phi\,\beta} \Big\} (x^s)^{\frac{\beta+1}{2}} \int_0^\infty y^{\frac{\beta-1}{2}} \exp\Big\{ -\frac{(x^s)^\beta}{\phi\,\beta(\beta-1)}\big( y^\beta - \beta\, y \big) \Big\}\, \mathrm{d}y. \;}$$

**Numerical evaluation**    This boxed expression is the normalizer used in the likelihood score. We do not introduce separate closed-form normalizers for special limiting distributions; in our experiments the Cressie–Read parameter is selected over a positive $\beta$-range, and the remaining $y$-integral is computed numerically (e.g. via Gauss–Laguerre quadrature or series expansions).

## F. Extensions to PAC-Bayes Bounds

Recent PAC-Bayes methods using data-dependent priors and optimization have produced non-vacuous generalization bounds for deep models. We outline two directions in which our Tighter-VR formulation can be extended to yield non-vacuous, data-dependent guarantees at large scale.

- **Ohnishi and Honorio (Ohnishi & Honorio, 2021)** use the classical variational representation of $f$- and $\alpha$-divergences to derive PAC-Bayes bounds, but these can be loose for over-parameterized networks. By injecting our Tighter-VR transform $\tau$ and restricting critics to a structured family (e.g., parameterized density ratios or small neural nets), we obtain a strictly tighter discrepancy measure on the same divergence. Substituting this improved estimate into any PAC-Bayes change-of-measure inequality directly reduces the complexity penalty—without altering the proof—and yields non-vacuous generalization guarantees for large-scale deep models.

- Prior work has already applied PAC-Bayes analyses to unsupervised domain adaptation—for example, Germain et al. (Germain et al., 2016) introduced a Donsker-Varadhan KL penalty, but in high-dimensional, over-parameterized regimes this term often becomes vacuous. We replace that KL penalty with our **Tighter-VR estimate of a general $f$-divergence**. By using stronger change-of-measure inequalities and restricting critics to valid density-ratio models, we obtain strictly smaller divergence penalties. This simple swap preserves the original proof structure while yielding **data-dependent, non-vacuous** UDA guarantees that improve on both classical KL-based PAC-Bayes and Donsker-Varadhan bounds.

## G. Algorithm Details

### G.1. Algorithm Overview

Figure 2 illustrates our UDA architecture. A shared feature encoder

$$h_{\text{rep}} : \mathcal{X} \rightarrow \mathcal{Z}$$

maps source inputs $x^s$ and target inputs $x^t$ into a common latent space. Two classifier heads—primary $h_{\text{cls}} : \mathcal{Z} \rightarrow \mathcal{Y}$ and auxiliary $h'_{\text{cls}} : \mathcal{Z} \rightarrow \mathcal{Y}$—then produce hypotheses

$$h = h_{\text{cls}} \circ h_{\text{rep}}, \quad h' = h'_{\text{cls}} \circ h_{\text{rep}}.$$

At each training iteration, we sample a mini-batch of $m$ labeled source examples $\{(x_i^s, y_i^s)\}_{i=1}^m$ and $m$ unlabeled target examples $\{x_j^t\}_{j=1}^m$. We compute the transformed source classification loss

$$\mathcal{L}_{\text{cls}} = \frac{1}{m} \sum_{i=1}^m \tau \circ \ell\big(h(x_i^s),\, y_i^s\big),$$

and estimate the tighter-VR discrepancy [1]

$$\tilde{d}_{\hat{\mu},\hat{\nu}}^\tau(h, h') = \mathbb{E}_{\hat{\nu}}\big[\tau\big(\ell(h(x), h'(x))\big)\big] - \mathbb{E}_{\hat{\mu}}\big[f^*\big(\tau(\ell(h(x), h'(x)))\big)\big],$$

where $\tau$ is a learnable transform (e.g. affine by default).

The combined loss

$$L = \mathcal{L}_{\text{cls}} + \eta\, \tilde{d}_{\hat{\mu},\hat{\nu}}^\tau(h, h')$$

is then optimized via an alternating minimax schedule (see Algorithm 1):

1. *Minimize $L$* with respect to $\big(h_{\text{rep}},\, h_{\text{cls}},\, \tau\big)$ by gradient descent.

---

**Algorithm 1** Training Adaptive $f$-Divergence UDA

---

**Require:** Labeled source set $\mathcal{S} = \{(x_i^s, y_i^s)\}_{i=1}^{n_s}$, unlabeled target set $\mathcal{T} = \{x_j^t\}_{j=1}^{n_t}$.
1:   feature encoder $h_{\mathrm{rep}}$, classifiers $h_{\mathrm{cls}}, h'_{\mathrm{cls}}$.
2:   transform parameters $(a, b)$ for $\tau(x) = a\,x + b$, trade-off $\eta > 0$, epochs $E$, batch size $m$.
     **Ensure:** Trained $h_{\mathrm{rep}}, h_{\mathrm{cls}}, h'_{\mathrm{cls}}$ and $(a, b)$.
3:   Initialize $h_{\mathrm{rep}}$ (ImageNet-pretrained or random), $h_{\mathrm{cls}}, h'_{\mathrm{cls}}$ randomly.
4:   Initialize $\tau$ (e.g., $(a, b) \leftarrow (1, 0)$).
5:   **for** $e = 1$ **to** $E$ **do**
6:      **for** each mini-batch $B_s \subset \mathcal{S}, B_t \subset \mathcal{T}$ with $|B_s| = |B_t| = m$ **do**
7:         **Source classification loss:** $\mathcal{L}_{\mathrm{cls}} \leftarrow \frac{1}{m} \sum\limits_{(x^s, y^s) \in B_s} \tau\Big(\ell\big(h_{\mathrm{cls}}(h_{\mathrm{rep}}(x^s)), y^s\big)\Big).$
8:         **Tight variational divergence estimate:** compute $\tilde{d}_{\hat{\mu}, \hat{\nu}}^{\tau}(h, h')$ as in Eq. (13).
9:         **Combined objective:** $L \leftarrow \mathcal{L}_{\mathrm{cls}} + \eta\,\tilde{d}_{\hat{\mu}, \hat{\nu}}^{\tau}(h, h')$.
10:        Update $(h_{\mathrm{rep}}, h_{\mathrm{cls}}, \tau)$ by one step of gradient *descent* on $L$.
11:        Update $h'_{\mathrm{cls}}$ by one step of gradient *ascent* on $\tilde{d}_{\hat{\mu}, \hat{\nu}}^{\tau}(h, h')$.
12:     **end for**
13:  **end for**

---

2. *Maximize $\tilde{d}_{\hat{\mu}, \hat{\nu}}^{\tau}(h, h')$ with respect to $h'_{\mathrm{cls}}$ by gradient ascent.*

We apply standard stabilization techniques—gradient reversal, weight decay, and learning-rate warm-up—to ensure convergence. By jointly learning the encoder, classifiers, and the divergence transform $\tau$, our algorithm dynamically adapts both the representation alignment and the choice of $f$-divergence in an end-to-end fashion.

### G.2. Parametric Transform Families.

In practice we choose the scalar map $\tau$ from four simple, monotone families and use them as *empirical controls/ablations* (our theory does not rely on any single choice):

$$\tau(x) = \begin{cases} a\,x + b, & \text{(Affine)} \\ \big(|x| + \varepsilon\big)^c, \ \varepsilon = 10^{-6}, & \text{(Power)} \\ \exp(\gamma x), & \text{(Exponential)} \\ \sigma(\gamma x) = \frac{1}{1 + e^{-\gamma x}}, & \text{(Sigmoid)} \end{cases} \tag{37}$$

with learnable $a, b, \gamma, c$ (init: $a{=}1, b{=}0, \gamma{=}1, c{=}1$). The small $\varepsilon$ prevents flat gradients near $x{=}0$. During training, the chosen $\tau$ is updated by backprop together with model parameters.

*Lipschitz note (for readers concerned with assumptions).* Only some choices are globally $L$-Lipschitz on $\mathbb{R}$: Affine is global with $L{=}|a|$, and Sigmoid is global with $L{=}|\gamma|/4$. Power is global only when $c \leq 1$ (then $L{=}|c|\varepsilon^{c-1}$); for $c > 1$ it is not global but is Lipschitz on bounded inputs $x \in [-B, B]$ with $L{=}|c|(B{+}\varepsilon)^{c-1}$. Exponential is not global; on $[-B, B]$ it admits $L{=}|\gamma|e^{|\gamma|B}$. We keep the Power ($c > 1$) and Exponential forms *solely as practical ablations* to probe shape flexibility under the standard bounded-feature preprocessing used in our experiments; they are not invoked when a global Lipschitz constant is required by theory.

## H. Experimental Details

### H.1. Experimental Setup

**Codebase and Reproducibility.** Our implementation is based on the open-source Transfer Learning Library, accessible at https://github.com/thuml/Transfer-Learning-Library, a unified framework for transfer learning and domain adaptation (Jiang et al., 2022). In particular, we extend the implementation of $f$-DAL (Acuna et al., 2021) within this library. We retain the overall architecture and training protocol from $f$-DAL, and adapt it to incorporate our proposed

---

[1]If the adaptive $f$-divergence is used, replace $\tilde{d}_{\hat{\mu}, \hat{\nu}}^{\tau}(h, h')$ with the empirical analogue of Eq. (12).

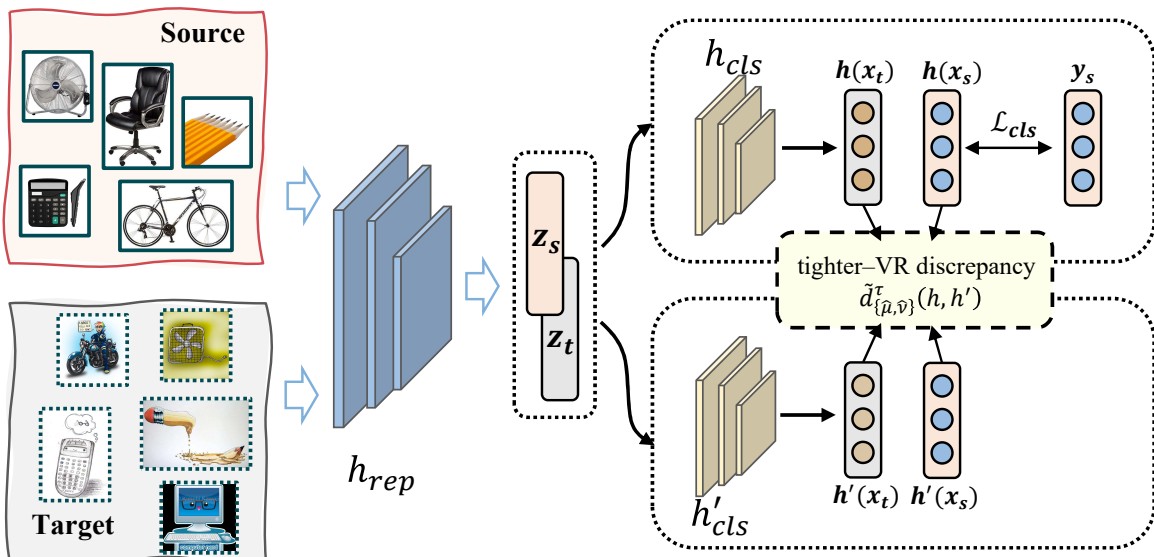

*Figure 2.* Overview of our model. A shared encoder $h_{\text{rep}}$ maps source inputs $x_s$ and target inputs $x_t$ into feature vectors $z_s$ and $z_t$. Two classifiers—primary $h_{\text{cls}}$ and auxiliary $h'_{\text{cls}}$—take these features to produce hypotheses $h$ and $h'$ for both domains. During training, we jointly minimize the source classification loss $\mathcal{L}_{\text{cls}}(h)$ on labeled source data and maximize the tighter variational-representation discrepancy $\tilde{d}^\tau_{\hat{\mu},\hat{\nu}}(h, h')$ between source and target, where the parametric transform $\tau$ is learned end-to-end to ensure a valid lower bound on the chosen $f$-divergence.

tighter discrepancy measure, learnable transformation module, and adaptive divergence selection. All experiments were conducted on NVIDIA A40 GPUs (48GB).

**Experiment Settings and Hyperparameter Tuning.** For the Digits benchmark, we employ a LeNet backbone; for the Office-31 and Office-Home tasks, we use ResNet-50 pretrained on ImageNet (Deng et al., 2009); and for the large-scale VisDA-2017 benchmark, we adopt pretrained ResNet-101. All main ResNet/LeNet experiments share the same optimization protocol: mini-batch size 32, SGD with initial learning rate $10^{-3}$ for Digits (and $4 \times 10^{-3}$ for Office), weight decay $5 \times 10^{-4}$, and 50 total epochs. Reported accuracies are averaged over three independent random seeds unless otherwise stated.

The $f$-DD baseline in Tables 1, 2, and 3 is implemented with the Pearson $\chi^2$ divergence ($\chi^2$-DD) to allow a direct comparison under the same divergence family. Prior work on $\chi^2$-DD (Wang & Mao, 2024) observed that tuning its fixed transformation often yields negligible gains; by contrast, our jointly learned $\tau$ improves performance even when the underlying divergence is held fixed.

Our hyperparameter settings and training schedule closely follow the Transfer Learning Library (Jiang et al., 2022). In particular, we sweep the trade-off coefficient $\eta$ over $\{0.1, 0.3, 0.5, \ldots, 5.0\}$. The initial values for the learnable transform $\tau$ are as in Eq. (37): for the affine form $a = 1, b = 0$, for the power form $c = 1$, and for the exponential/sigmoid forms $\gamma = 1$.

## I. Additional Experiments

### I.1. Additional Results on Different Transformations

All experiments in this ablation use the Pearson $\chi^2$ divergence and vary only the choice of the parametric transform $\tau$. We compare four families—power, exponential, sigmoid, and affine—on the standard Office-31 and Office-Home benchmarks. Table 8 reports six transfer tasks on Office-31, while Table 9 covers all twelve Office-Home pairs. In both settings the learnable *affine* map $\tau(x) = a\,x + b$ consistently achieves the highest average accuracy (90.4 % on Office-31; 69.9 % on Office-Home). Power and exponential transforms yield competitive results (within 0.6 pp), and sigmoid falls slightly behind. These results indicate that, when using $\chi^2$ divergence, jointly learning a simple affine re-scaling of the surrogate loss provides the largest practical benefit.

*Table 8.* Accuracy (%) on Office-31 (mean ± std).

| Transformation | $\mathbf{A \to W}$ | $\mathbf{D \to W}$ | $\mathbf{W \to D}$ | $\mathbf{A \to D}$ | $\mathbf{D \to A}$ | $\mathbf{W \to A}$ | **Avg** |
|---|---|---|---|---|---|---|---|
| Power | $95.5 \pm 0.1$ | $98.7 \pm 0.4$ | $100 \pm 0$ | $94.8 \pm 0.4$ | $74.6 \pm 0.5$ | $75.5 \pm 0.6$ | 89.8 |
| Exponential | $95.0 \pm 0.3$ | $98.9 \pm 0.2$ | $100 \pm 0$ | $94.8 \pm 0.2$ | $74.8 \pm 0.2$ | $75.3 \pm 0.2$ | 89.8 |
| Sigmoid | $94.9 \pm 0.3$ | $99.0 \pm 0.3$ | $100 \pm 0$ | $96.2 \pm 0.3$ | $74.0 \pm 0.3$ | $75.1 \pm 0.2$ | 89.9 |
| Affine | $95.5 \pm 0.3$ | $98.9 \pm 0.1$ | $100 \pm 0$ | $95.8 \pm 0.2$ | $75.5 \pm 0.5$ | $76.4 \pm 0.2$ | 90.4 |

*Table 9.* Accuracy (%) on Office-Home.

| Transformation | $\mathbf{Ar \to Cl}$ | $\mathbf{Ar \to Pr}$ | $\mathbf{Ar \to Rw}$ | $\mathbf{Cl \to Ar}$ | $\mathbf{Cl \to Pr}$ | $\mathbf{Cl \to Rw}$ | $\mathbf{Pr \to Ar}$ | $\mathbf{Pr \to Cl}$ | $\mathbf{Pr \to Rw}$ | $\mathbf{Rw \to Ar}$ | $\mathbf{Rw \to Cl}$ | $\mathbf{Rw \to Pr}$ | **Avg** |
|---|---|---|---|---|---|---|---|---|---|---|---|---|---|
| Power | 55.4 | 72.8 | 78.1 | 62.9 | 73.2 | 73.5 | 62.3 | 53.4 | 81.1 | 74.8 | 61.1 | 84.4 | 69.4 |
| Exponential | 55.6 | 73.6 | 78.5 | 62.6 | 73.3 | 73.1 | 63.5 | 53.6 | 81.4 | 74.6 | 61.7 | 84.3 | 69.7 |
| Sigmoid | 55.3 | 73.3 | 78.3 | 62.4 | 73.5 | 72.7 | 62.6 | 53.6 | 81.0 | 74.9 | 61.5 | 84.3 | 69.5 |
| Affine | 55.7 | 74.2 | 79.1 | 62.8 | 73.8 | 73.8 | 63.1 | 54.0 | 81.1 | 75.2 | 61.5 | 84.3 | 69.9 |

## I.2. Additional Results on Divergence Selection.

Table 10 reports results on three datasets for several fixed divergences, all with the same affine $\tau$. We compare

- **KL-D**: the forward KL divergence $\mathrm{KL}(P\|Q)$,

- $\boldsymbol{\chi^2}$**-D**: the Pearson $\chi^2$ divergence $\mathbb{E}_Q[(\frac{dP}{dQ} - 1)^2]$,

- **RKL**: the *reverse* KL, $\mathrm{KL}(Q\|P)$,

- **Neyman**: the "Neyman" $\chi^2$ divergence $\mathbb{E}_P[(\frac{dQ}{dP} - 1)^2]$.

- **JS-D**: the Jensen–Shannon divergence $\mathrm{JS}(P\|Q) = \frac{1}{2}\mathrm{KL}(P\|M) + \frac{1}{2}\mathrm{KL}(Q\|M), \quad M = \frac{1}{2}(P + Q)$, which symmetrises and smooths the KL divergence.

- **Jeffreys-D**: the Jeffreys divergence $J(P,Q) = D(P\|Q) + D(Q\|P)$, the symmetric counterpart of KL used by $f$-DD (Jeffreys-DD). We also report Ours (Affine–Jeffreys-DD), which keeps our affine $\tau$ objective unchanged and only replaces the fixed divergence with $J(\cdot,\cdot)$ for a head-to-head comparison.

RKL and Neyman–$\chi^2$ perform poorly on Digits (10.8% and 74.5%), so we do not report them on Office-31 and Office-Home. In contrast, both forward KL and Pearson $\chi^2$ remain competitive, with Pearson $\chi^2$ giving the best results.

*Table 10.* Effect of fixed divergence choice on datasets (affine $\tau$ in all cases).

| Divergence | Office-31 | Office-Home | Digits |
|---|---|---|---|
| KL-D | 90.2 | 69.6 | 96.8 |
| $\chi^2$-D | 90.4 | 69.9 | 96.9 |
| JS-D | 89.9 | 69.7 | 96.7 |
| RKL ($\mathrm{KL}(Q\|P)$) | – | – | 10.8 |
| Neyman-$\chi^2$ | – | – | 74.5 |

Consequently, the Pearson $\chi^2$ divergence consistently outperforms the other fixed-divergence measures, and we therefore adopt it for the main fixed-divergence experiments.

**Adding Jeffreys as a symmetric baseline.** Prior work reports strong results with Jeffreys-DD (symmetric KL). To isolate the effect of the divergence *itself*, we keep our affine $\tau$ objective and training protocol unchanged, and simply swap the fixed divergence: (i) $f$-DD (Jeffreys-DD) and (ii) *Ours (Affine–Jeffreys-DD)*. This comparison is orthogonal to our contribution and serves as a robustness check of our method under a symmetric divergence.

*Table 11.* **Office-Home (part)** with Jeffreys: six transfers Ar→Cl, Ar→Rw, Cl→Pr, Pr→Ar, Pr→Rw, Rw→Cl. Numbers are target top-1 (%). Our affine method remains competitive and leads on average when paired with Jeffreys.

| Method | Ar→Cl | Ar→Rw | Cl→Pr | Pr→Ar | Pr→Rw | Rw→Cl |
|---|---|---|---|---|---|---|
| $f$-DD (Jeffreys-DD) | 55.5 | 79.5 | 73.8 | 63.9 | 81.3 | 61.6 |
| Ours (Affine-$\chi^2$-DD) | 55.7 | 79.1 | 73.8 | 63.1 | 81.1 | 61.5 |
| **Ours (Affine-Jeffreys-DD)** | **56.1** | **80.4** | **74.4** | **64.2** | **81.9** | **62.2** |

*Table 12.* **Office-31** with Jeffreys. Columns are A→W, D→W, W→D, A→D, D→A, W→A, and the mean. Our affine $\tau$ method combined with Jeffreys remains the best on average among Jeffreys-based variants.

| Method | A→W | D→W | W→D | A→D | D→A | W→A | Avg. |
|---|---|---|---|---|---|---|---|
| $f$-DD (Jeffreys-DD) | 94.9 | 99.1 | 100 | 95.9 | 76.0 | 74.6 | 90.1 |
| Ours (Affine-$\chi^2$-DD) | **95.5** | 98.9 | 100 | 95.8 | 75.5 | 76.4 | 90.4 |
| **Ours (Affine-Jeffreys-DD)** | 95.2 | **99.3** | **100** | **96.3** | **76.3** | 76.2 | **90.6** |

**Takeaway for divergence choice.** Jeffreys, as a symmetric counterpart of KL, is a strong fixed divergence; pairing it with our affine $\tau$ delivers additional gains over $f$-DD (Jeffreys-DD) on both Office-Home and Office-31. Pearson $\chi^2$ remains the default setting in the main experiments, while Jeffreys is reported here as a complementary symmetric option that gives strong results on the reported Office subsets.

### I.3. Comparison with Optimal Transport Baselines

Since Wasserstein and optimal transport objectives are natural discrepancy-based competitors for UDA, we also compare against representative OT-based baselines. Table 13 provides a cross-family reference point under common UDA benchmarks.

*Table 13.* Comparison with representative optimal-transport baselines. Numbers are target-domain top-1 accuracy (%).

| Method | Office-31 | Office-Home | VisDA-2017 |
|---|---|---|---|
| JDOT (Courty et al., 2017) | 83.5 | 60.5 | 74.7 |
| DeepJDOT (Damodaran et al., 2018) | 86.2 | 50.6 | 77.4 |
| MLOT (Kerdoncuff et al., 2020) | 87.8 | 66.2 | 73.0 |
| **Ours (Affine-$\chi^2$)** | **90.4** | **69.9** | **78.9** |

Our method remains competitive with these OT-based approaches while using the same tighter variational $f$-divergence objective studied throughout the paper. The result supports the claim that a learned transform and adaptive divergence choice can provide a strong alignment signal beyond comparisons restricted to prior $f$-divergence baselines.

### I.4. Architecture Generality with ViT (DeiT-S/16)

To assess whether our method depends on a particular convolutional backbone, we further replace ResNet-50 with a ViT backbone (DeiT-S/16) (Touvron et al., 2021). Unless otherwise noted, we keep the training protocol identical to the ResNet-50 setup: ImageNet-1k pretraining, input resolution $224 \times 224$, identical data augmentation and number of updates, AdamW with the same schedule, and the alignment head unchanged. We use the global representation (the [CLS] token) as $z$ and apply our objective on $z$ exactly as in the CNN case. This ensures that any performance differences stem from the architecture rather than training heuristics.

I.4.1. RESULTS ON OFFICE-HOME (3 TASKS) AND VISDA-2017

We report results on three representative Office-Home transfers (Ar→Cl, Ar→Pr, Ar→Rw) and on VisDA-2017 using DeiT-S/16. Table 14 summarizes mean $\pm$ std over $k$ runs (we use $k=5$ in our experiments).

Across all four settings, our method improves the DeiT-S/16 baseline and outperforms the strongest prior method under the same backbone. The gains are modest but consistent (about $+1.5\sim3.0$ pp), mirroring the trend observed with ResNet-50 and supporting the claim that our objective transfers across architectures.

*Table 14.* Accuracy (%) with DeiT-S/16 (mean $\pm$ std over 5 runs).

| Method | Ar→Cl | Ar→Pr | Ar→Rw | VisDA-2017 |
|---|---|---|---|---|
| DeiT-based | 61.8±0.2 | 79.5±0.3 | 84.3±0.3 | 73.2±0.4 |
| $f$-DAL (DeiT-S/16) | 66.9±0.3 | 82.4±0.3 | 85.7±0.3 | 84.7±0.4 |
| $f$-DD (DeiT-S/16) | 69.4±0.3 | 86.7±0.3 | 86.4±0.1 | 86.1±0.2 |
| **Ours (Affine-$\chi^2$, DeiT-S/16)** | **72.1**±0.1 | **88.3**±0.2 | **87.9**±0.2 | **89.1**±0.1 |

### I.4.2. RESULTS ON DOMAINNET C/P/R/S

To further test whether the proposed discrepancy remains useful on a larger and more heterogeneous benchmark, we evaluate a DeiT-S/16 backbone on a 12-task DomainNet subset using the Clipart, Painting, Real, and Sketch domains. All methods use the same backbone and protocol. For $f$-DD we use the fixed $\chi^2$ divergence, while our method uses adaptive Cressie–Read $\beta$ selection with an affine transform and freezes $\beta$ after 10 epochs.

*Table 15.* DomainNet C/P/R/S results with DeiT-S/16. Numbers are target-domain top-1 accuracy (%).

| Method | C→P | C→R | C→S | P→C | P→R | P→S | R→C | R→P | R→S | S→C | S→P | S→R | Avg |
|---|---|---|---|---|---|---|---|---|---|---|---|---|---|
| DANN (Ganin et al., 2016) | 48.3 | 52.0 | 59.2 | 47.3 | 62.1 | 46.7 | 65.2 | 63.4 | 62.1 | 46.2 | 36.7 | 40.3 | 52.5 |
| $f$-DAL (Acuna et al., 2021) | 51.2 | 57.3 | 62.1 | 50.2 | 64.7 | 50.3 | 68.5 | 67.2 | 64.9 | 48.7 | 37.9 | 44.7 | 55.6 |
| $f$-DD (Wang & Mao, 2024) | 52.7 | 58.2 | 62.7 | 51.7 | 65.4 | 52.2 | 69.7 | 68.8 | 65.7 | 50.2 | 38.6 | 46.3 | 56.9 |
| **Ours** | **54.1** | **60.3** | **64.2** | **53.6** | **67.5** | **54.1** | **71.2** | **70.6** | **68.0** | **51.9** | **40.2** | **48.8** | **58.7** |

Our method improves over the strongest baseline on all 12 transfer directions and reaches an average accuracy of 58.7%, suggesting that the learned discrepancy is not tied to the original ResNet-50 setting.

### I.4.3. BROADER COMPARISON WITH RECENT UDA SYSTEMS

We also compare with recent UDA systems under broader settings. This table is not a matched-protocol comparison because the methods use different backbones, pretraining sources, and training pipelines; it is included to contextualize our DeiT-S/16 results.

*Table 16.* Broader comparison with recent UDA references. Numbers are target-domain top-1 accuracy (%).

| Method | Backbone | Ar→Cl | Ar→Pr | Ar→Rw | VisDA-2017 |
|---|---|---|---|---|---|
| CDTrans-B (Xu et al., 2022) | DeiT-B/16 | 68.8 | 85.0 | 86.9 | 88.4 |
| TVT (Yang et al., 2023) | ViT-B/16 | 74.9 | 86.8 | 89.5 | 86.7 |
| PDA (Bai et al., 2024) | CLIP ViT-B/16 | 73.5 | 91.4 | 91.3 | 89.7 |
| SWG DAPL V2 (Westfechtel et al., 2025) | CLIP ViT-B/16 | 87.1 | 93.9 | 94.1 | 92.3 |
| **Ours** | DeiT-S/16 | 72.1 | 88.3 | 87.9 | 89.1 |

Under a lighter DeiT-S/16 backbone, our method is competitive with several transformer-based references, while CLIP-based pipelines remain stronger on some Office-Home tasks due to substantially different pretraining and prompt-adaptation components. We therefore use this table as a broader reference comparison, not as a controlled ranking.

### I.4.4. STATISTICAL ANALYSIS PROTOCOL AND SUMMARY

**Protocol.** We fix dataset splits and preprocessing across runs and vary only the random seed for initialization and data order. For each task $t$ and seed $i \in \{1, \ldots, k\}$, let $a_{t,i}^{\text{ours}}$ and $a_{t,i}^{\text{base}}$ denote accuracies of our method and a chosen comparator under the *same* DeiT backbone and training protocol. Define paired differences $d_{t,i} = a_{t,i}^{\text{ours}} - a_{t,i}^{\text{base}}$. We report: (i) mean $\bar{d}_t$ and standard deviation $s_{d,t}$; (ii) a two-sided paired $t$-test with statistic $t_t = \bar{d}_t/(s_{d,t}/\sqrt{k})$ (df = $k-1$), together with a 95% confidence interval $\bar{d}_t \pm q_{0.975} \, s_{d,t}/\sqrt{k}$ (where $q_{0.975}$ is the Student-$t$ quantile); (iii) Cohen's $d$ effect size $d_{\text{Cohen},t} = \bar{d}_t/s_{d,t}$; and (iv) a nonparametric Wilcoxon signed-rank test as a robustness check. When comparing on multiple tasks, we also report the macro-average $\bar{d} = \frac{1}{T}\sum_{t=1}^{T} \bar{d}_t$ and apply Benjamini–Hochberg correction to control the false discovery rate when repeated-run statistics are available.

All DeiT-S/16 experiments follow the same pretraining, resolution, augmentation, and training schedule as their ResNet-50 counterparts; the observed gains are consistent with the architecture-agnostic role of our objective.

### I.5. Additional Validation: Our Method with an Optional RK2 Optimizer

Our primary contribution is the affine-divergence based adaptation objective and model design. In this appendix, we *only* replace the default GRL/Euler updates with a second-order Runge-Kutta (RK2) step (Acuna et al., 2022) as an **optional optimizer**, to test the stability of our objective under more challenging training dynamics. All architectures, losses, and hyperparameters remain unchanged unless stated otherwise.

Adversarial updates form a coupled min-max system; explicit Euler (GRL) can exhibit rotational dynamics that slow convergence. RK2 uses a mid-point forecast before the final update, which typically damps such rotations. This modification is *optimizer-level and orthogonal* to our objective: it does not alter our loss, regularizers, or architectures.

Let $u$ collect the minimization blocks (feature extractor + classifier) and $v$ be the maximization block (domain discriminator). Denote $G_u = \nabla_u \mathcal{L}$ and $G_v = \nabla_v \mathcal{L}$ for our **unchanged** loss $\mathcal{L}$. One RK2 step (mid-point) with stepsizes $\eta_u, \eta_v$ is:

$$k_u^{(1)} = -G_u(u_t, v_t), \quad k_v^{(1)} = G_v(u_t, v_t), \quad \tilde{u} = u_t + \tfrac{\eta_u}{2} k_u^{(1)}, \ \tilde{v} = v_t + \tfrac{\eta_v}{2} k_v^{(1)},$$

$$k_u^{(2)} = -G_u(\tilde{u}, \tilde{v}), \quad k_v^{(2)} = G_v(\tilde{u}, \tilde{v}), \quad u_{t+1} = u_t + \eta_u k_u^{(2)}, \ v_{t+1} = v_t + \eta_v k_v^{(2)}.$$

This adds one extra gradient evaluation per step and can reduce the number of iterations needed to reach the target accuracy.

Using RK2 on top of *our* affine divergence model improves over strong $f$-divergence baselines without changing our objective. The summary is below:

*Table 17.* **Target-domain top-1 accuracy (%).** RK2 is an *optional, optimizer-level* enhancement that *does not* change our loss or architecture. It acts as a stability check for our objective and brings up to $+3.3$ pp over $f$-DD on VisDA-2017. *Office-Home (part) includes the six transfers: Ar→Cl, Ar→Rw, Cl→Pr, Pr→Ar, Pr→Rw, Rw→Cl.*

| Method | Office-31 | Office-Home (part) | Digits | VisDA-2017 |
|---|---|---|---|---|
| $f$-DAL | 89.5 | 67.8 | 96.3 | 74.3 |
| $f$-DD ($\chi^2$) | 89.9 | 68.8 | 96.4 | 76.2 |
| **Ours (Affine + RK2)** | **90.8** | **70.7** | **97.3** | **79.5** |

We keep all hyperparameters as in the main text; RK2 tolerates a slightly larger stepsize and requires $\sim 1.7\times$ per-step compute, but typically converges in fewer iterations. We found mild decoupling between generator and discriminator stepsizes helpful (e.g., $\eta_v \in [0.5, 1.0]\eta_u$), though our objective remained stable across a wide range.

These results *support* (rather than redefine) our contribution: the affine-divergence objective remains stable across optimizers. RK2 provides an optimizer-level variant of the same objective.

### I.6. Adaptive Cressie–Read $\beta$ Selection on Office-Home and VisDA-2017

Digits is near-saturation, so we further test the Cressie–Read divergence parameter $\beta$ on harder Office-Home transfer tasks and VisDA-2017. Our method, architecture, and all hyperparameters remain unchanged; only $\beta$ is updated by a lightweight rule and then optionally frozen. We compare fixed $\beta$, fixed $\beta$ with the learnable transform, fully adaptive $\beta$, and adaptive $\beta$ with freezing after 10 epochs. Metrics are target top-1 accuracy (%).

*Table 18.* **Adaptive Cressie–Read $\beta$ selection on Office-Home and VisDA-2017.** Adaptive $f$-divergence selection gives consistent gains over fixed $\beta$ with learnable $\tau$; freezing after 10 epochs is a stable trade-off.

| Method | VisDA-2017 | Ar→Rw | Pr→Cl | Ar→Cl |
|---|---|---|---|---|
| Fixed $\beta$ | 74.6 | 78.1 | 53.3 | 54.7 |
| Fixed $\beta$ with learnable $\tau$ | 78.9 | 79.1 | 54.0 | 55.7 |
| Fully Adaptive-$\beta$ | **79.8** | 79.6 | 54.9 | **56.3** |
| Adaptive-$\beta$ (freeze after 10ep) | 79.7 | **79.7** | **55.1** | 56.2 |

Relative to fixed $\beta$ with learnable $\tau$, adaptive selection improves Ar→Rw from 79.1 to 79.7, Pr→Cl from 54.0 to 55.1, Ar→Cl from 55.7 to 56.3, and VisDA-2017 from 78.9 to 79.8. These results support the claim that the adaptive divergence selector contributes beyond the learned transform alone.

### I.7. Feature Visualization via t-SNE

Figure 3 shows t-SNE (Van der Maaten & Hinton, 2008) embeddings for the transfer task MNIST→USPS using four adaptation methods:

- DANN shows large source/target overlap with no clear class separation.

- $f$-DAL clusters arise but remain partially misaligned across domains.

- A learnable *sigmoid* transform tightens clusters and improves alignment.

- The learnable *affine* transform produces compact, well-separated clusters with strong source–target colocation.

These visualizations are consistent with the quantitative ablation results (Table 4): the tighter discrepancy measure combined with an adaptive transform yields more discriminative, domain-invariant features.

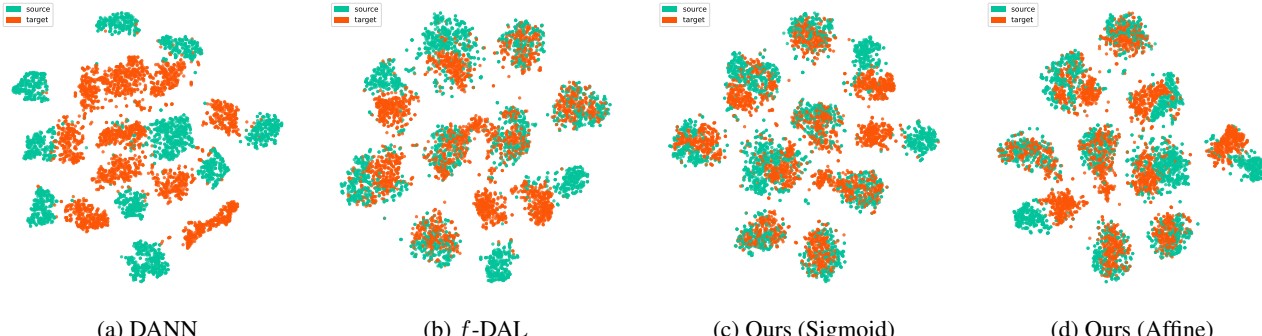

(a) DANN        (b) $f$-DAL        (c) Ours (Sigmoid)        (d) Ours (Affine)

*Figure 3.* t-SNE visualizations of learned feature spaces under different adaptation methods.

