# OpenReview forum: "Domain Adaptation with Adaptive $f$-Divergence: Tighter Variational Representation and Generalization Bounds"
_ICML.cc/2026/Conference — ICML 2026 regular_

### Official Review · Reviewer_bVaN · 2026-03-03

**Soundness:** 3
**Presentation:** 2
**Significance:** 3
**Originality:** 2
**Overall Recommendation:** 4
**Confidence:** 3

**Summary:**

This paper starts from the measurement of domain discrepancy, which is one crucial part in domain adaptation. They argue that the conventional f-divergence is suboptimal, and propose to insert a learnable, monotone lipschitz transform to tighten the variational lower bound. Additionally, it selects the divergence family adaptively via a likelihood criterion. It provide theoretical bound for the proposed method, and provide finite-sample guarantee. Experiment results prove the effectiveness of the proposed method.

**Compliance With Llm Reviewing Policy:**

Affirmed.

**Final Justification:**

After reading the rebuttal, I keep my score

**Key Questions For Authors:**

See weakness. I think the authors can have more discussions on
1. experiments, comparison with more recent methods or some more analysis experiments.
2. novelty, compare your method with previous method in detail and list your novelty one by one. it will also be beneficial to the community.

**Limitations:**

yes.

**Strengths And Weaknesses:**

Strengths:
1. This paper focuses on a very fundamental problem in domain adaptation, the measurement of domain discrepancy.
2. This paper provides detailed theoretical insights, it is very necessary when we introduce a metric to quantify the domain discrepancy.
3. The whole paper is in a very clear organization.
4. It is good that the authors have discussions on the limitations and impact of the work, which will be beneficial to the society.

Weaknesses:
1. Experiments: The baseline methods in the experiment section is somewhat out-of-dated. The authors can add more recent baseline methods, especially in 2023-2025. I know that it is a theoretical paper, so I do not pursue that the proposed method can outperform all of them (reach state-of-the-art performance), but it will be more convincing and also interesting to see what happened when the proposed method is plugged to existing method.
2. Novelty: I think the authors can have more discussions about the novelty of the proposed method. Being the first to use an existing metric in domain adaptation in fact is novel, but the novelty is limited. The authors can have more discussions on how you design your method, make your idea shine.

---

> ### Author Rebuttal · Authors · 2026-03-30
>
> ### W1 & Q1. Experiments and Comparison with Recent Methods
>
> We thank the reviewer for the constructive suggestion. We agree that broader comparisons and additional analysis would strengthen the paper. In the main submission, we intentionally adopt the canonical ResNet-50 protocol on standard UDA benchmarks and compare under a common setting, so that the effect of the proposed transformed discrepancy can be isolated in a protocol-matched way.
>
> To address the reviewer’s concern about newer methods, we additionally report recent transformer/VLM-based UDA references together with our DeiT-S/16 result from Table 13:
>
>
> | Method | Backbone | Ar→Cl | Ar→Pr | Ar→Rw | VisDA-2017 Avg |
> | --- | --- | ---: | ---: | ---: | ---: |
> | CDTrans-B [1]| DeiT-B/16 | 68.8 | 85.0 | 86.9 | 88.4 |
> | TVT [2]| ViT-B/16 | 74.9 | 86.8 | 89.5 | 86.7 |
> | PDA [3]| CLIP ViT-B/16 | 73.5 | 91.4 | 91.3 | 89.7 |
> | SWG DAPL V2 [4]| CLIP ViT-B/16 | 87.1 | 93.9 | 94.1 | 92.3 |
> | Ours | DeiT-S/16 | 72.1 | 88.3 | 87.9 | 89.1 |
>
> Under a lighter DeiT-S/16 backbone, our method already outperforms CDTrans-B on all four tasks and exceeds TVT on Ar$\rightarrow$Pr and VisDA-2017, while remaining below some stronger CLIP-based pipelines such as PDA and SWG DAPL V2. We report these as broader modern-reference baselines rather than strictly matched protocol baselines, since some of them use larger ViT-B/16 backbones and stronger CLIP pretraining.
>
> We also add a DeiT-S/16-based DomainNet subset evaluation (C/P/R/S, 12 transfer tasks) in the rebuttal **(Reviewer feD4 W1&Q3)**. This larger and more heterogeneous benchmark yields 58.7 average accuracy, confirming that the method is not tied to the original ResNet-50 setting. We will clarify this evaluation rationale and expand the discussion of newer methods in the revision.
>
>
> ### W2 & Q2. Novelty Clarification
>
> We thank the reviewer for this helpful suggestion. We agree that the novelty should be compared to the closest prior UDA methods more explicitly and listed point-by-point.
>
> Our method is most directly related to $f$-DAL and $f$-DD, but differs from them in the following aspects.
>
> **Compared with $f$-DAL:**
> $f$-DAL uses the classical Fenchel variational representation in UDA, which corresponds to the identity transform in our framework. Our method replaces this fixed form by a learnable monotone Lipschitz transform $\tau$, optimized jointly with the adaptation model. Thus, $f$-DAL is recovered as the special case $\tau(x)=x$.
>
> **Compared with $f$-DD:**
> $f$-DD improves over $f$-DAL by introducing a fixed scaling transform inside the variational form. However, both the transform and the divergence family remain fixed. Our method generalizes this in two ways: (i) the transform $\tau$ is learned from data rather than fixed a priori, and (ii) the divergence family is selected adaptively rather than pre-specified. Hence $f$-DD is recovered as a restricted special case of our framework.
>
> Therefore, the novelty of our paper is not merely applying an existing metric to UDA. The novelty is:
> (i) a **new adaptive UDA discrepancy** based on a learnable tighter variational transform $\tau$;
> (ii) a **data-driven divergence-family selection mechanism** instead of a fixed divergence choice; and
> (iii) the **corresponding UDA-specific theory**, including the target-risk decomposition and finite-sample guarantees for this transformed/adaptive discrepancy.
>
> We already provide a concise comparison with $f$-DAL/$f$-DD in **Appendix B (Table 7)**, and we will move this comparison more explicitly into the introduction and related-work discussion in the revision so that the novelty is easier to identify.
>
> **Reference**
>
> [1] CDTrans: Cross-domain Transformer for Unsupervised Domain Adaptation. ICLR 2022
>
> [2] TVT: Transferable Vision Transformer for Unsupervised Domain Adaptation. WACV 2023
>
> [3] Prompt-based Distribution Alignment for Unsupervised Domain Adaptation. AAAI2024
>
> [4] Combining Inherent Knowledge of Vision-Language Models with Unsupervised Domain Adaptation through Strong-Weak Guidance. WACV 2025

---

> > ### Author Rebuttal · Reviewer_bVaN · 2026-04-06
> >
> > I keep my score

---

> > > ### Author Response · Authors · 2026-04-06
> > >
> > > Thank you for the helpful follow-up. We are very glad that our rebuttal addressed your concerns. We will incorporate the relevant clarifications clearly in the final version. Thank you again for your time and feedback.

---

### Official Review · Reviewer_nmGf · 2026-03-10

**Soundness:** 3
**Presentation:** 2
**Significance:** 3
**Originality:** 3
**Overall Recommendation:** 4
**Confidence:** 2

**Summary:**

This paper studies unsupervised domain adaptation (UDA) with adaptive $\(f\)$-divergence estimation. It proposes (1) a tighter variational representation by introducing a learnable monotone Lipschitz transform $\(\tau\)$, and (2) a likelihood-based method to adaptively select the divergence family from data instead of fixing it a priori. The paper also derives target-risk and finite-sample generalization bounds. Experiments on Office-31, Office-Home, Digits, and VisDA-2017 show consistent improvements over prior $\(f\)$-divergence-based UDA methods such as f-DAL and f-DD.

**Compliance With Llm Reviewing Policy:**

Affirmed.

**Final Justification:**

I maintain my initial review.

**Key Questions For Authors:**

1) How much of the final gain comes from the learnable transform τ versus the adaptive divergence selection mechanism?

2) Can the authors clarify how the likelihood-based divergence selection in Eq. (11) and Eq. (12) is implemented in practice?

3) Why is adaptive divergence selection only limitedly evaluated in the main paper, despite being a main contribution?

4) What is the computational overhead relative to f-DAL and f-DD?

**Limitations:**

yes

**Strengths And Weaknesses:**

### Strengths

1. Well-motivated problem setting. The paper addresses a meaningful limitation of existing $\(f\)$-divergence-based UDA methods, namely the use of a fixed discrepancy measure across different domain shifts.

2. Technically coherent contribution. The combination of a learnable transform $\(\tau\)$ for tighter variational representation and adaptive divergence selection is conceptually clean. The framework also subsumes prior fixed-divergence approaches as special cases.

3. Strong theoretical coverage. The paper provides a population target-risk bound and multiple finite-sample generalization bounds (Rademacher, covering number, VC dimension). This gives the work a solid theoretical foundation.

4. Clear connection to prior work. The paper positions itself naturally relative to adversarial UDA, $\(f\)$-divergence-based adaptation, and tighter variational representations.
---

### Weaknesses

1. Adaptive divergence selection is not validated broadly enough in the main paper. The adaptive $\(\beta\)$-selection results are mainly demonstrated on Digits. Since adaptive divergence selection is one of the paper’s central claims, stronger evidence on larger benchmarks such as Office-Home or VisDA would strengthen the paper significantly.

3. Practical details of the likelihood-based selection are somewhat hard to follow. The construction through Tweedie-linked likelihood and the mapping between $\(\beta\)$- and $\(\alpha\)$-divergences is mathematically interesting, but the practical implementation details are not fully transparent from the main paper.

4. Theory is extensive but may exceed what is empirically substantiated. The paper presents many generalization bounds, but the connection between these results and the observed empirical behavior is not always sharply demonstrated. Some of the theory may feel standard rather than directly explanatory of the adaptive mechanism.

5. Limited scope of evaluation. The experiments focus on closed-set image classification benchmarks. It remains unclear how well the approach would transfer to more challenging settings such as partial/open-set DA, source-free DA, or larger-scale modern adaptation tasks.

---

> ### Author Rebuttal · Authors · 2026-03-30
>
> We thank the reviewer for the constructive feedback and helpful questions.
> ### W1/Q1/Q3. Adaptive Divergence Selection
> In the submitted manuscript, we intentionally separate the effects of the learnable transform $\tau$ and adaptive divergence selection. In the main benchmark tables, we fix the divergence family to Pearson-$\chi^2$ so that the gain from the tighter discrepancy and learnable transform can be evaluated in a controlled comparison against $f$-DAL/$f$-DD. Adaptive-$\beta$ is then studied separately to isolate its contribution.
>
> Beyond the Digits results in the main text, Appendix I.5 reports additional adaptive-$\beta$ results on Office-Home and VisDA-2017:
>
> | Method | VisDA-2017 | Ar->Rw | Pr->Cl | Ar->Cl |
> | --- | --- | --- | --- | --- |
> | Fixed $\tau = x$ | 74.6 | 78.1 | 53.3 | 54.7 |
> | Fixed $\beta = 2.0$ with learnable $\tau$ | 78.9 | 79.1 | 54.0 | 55.7 |
> | Fully Adaptive-$\beta$ | 79.8 | 79.6 | 54.9 | 56.3 |
> | Adaptive-$\beta$ (freeze after 10 ep) | 79.7 | 79.7 | 55.1 | 56.2 |
>
> Relative to fixed $\beta$ with learnable $\tau$, adaptive selection improves Ar$\rightarrow$Rw from 79.1 to 79.7, Pr$\rightarrow$Cl from 54.0 to 55.1, Ar$\rightarrow$Cl from 55.7 to 56.3/56.2, and VisDA-2017 from 78.9 to 79.8/79.7. This broader evidence will be made more explicit in the revision.
> ### W2 & Q2. Practical Details of Likelihood
> We agree that the practical implementation of the likelihood-based selector was not sufficiently explicit in the main text. Eq. (11) defines the likelihood-based selection objective. In implementation, we optimize its empirical mini-batch counterpart jointly with the main network via online SGD updates over the scalar parameters $(\beta,\phi)$, rather than introducing a separate closed-form inner optimization or repeated search over candidate divergence parameters. At each iteration, the current mini-batches define the empirical likelihood objective, the current $\beta$ instantiates the discrepancy term, and the likelihood term is added to the total loss with a small weight to guide and stabilize the update. Both variables are parameterized through positive transforms and constrained to valid ranges for numerical stability. This avoids grid search and adds no extra encoder/classifier/selector network.
>
> Eq. (12) is used only to map the selected $\beta$ to the corresponding $\alpha$-divergence; it is not a separate optimization stage. We will make this procedure explicit in the revision.
> ### W3. Theory-Empirics Connection
> We agree that the theory-to-empirics connection should be stated more sharply. Our main theoretical contribution is not the finite-sample machinery itself, which is standard, but the transformed tighter discrepancy and the induced target-risk decomposition under adaptive divergence modeling.
>
> The role of the theory is therefore to justify the surrogate/reduction and show statistical control of the transformed objective, rather than to tightly predict gains on every benchmark. The empirical results provide the complementary evidence that this objective improves performance in practice: the learnable transform $\tau$ already improves over $f$-DAL/$f$-DD under a fixed divergence family, and adaptive divergence selection yields further gains when enabled. We will make this division of roles more explicit in the revision.
> ### W4. Limited Scope of Evaluation
> We agree that the current evaluation is limited to standard closed-set UDA. To address the concern about larger-scale modern adaptation tasks, we additionally include a DeiT-S/16 evaluation on a 12-task DomainNet subset (C/P/R/S), where our method achieves the best average accuracy, see **reviewer feD4 W1&Q3** for details.
>
> For other DA variants, we would distinguish the scope more carefully. In partial DA, our transformed discrepancy and adaptive divergence selection could in principle be integrated into existing pipelines, but additional source-class reweighting/filtering is still needed to handle source-private classes. In open-set DA, the method is not a complete solution as is, since explicit unknown-class handling is required. In source-free DA, the current formulation is not directly applicable because discrepancy estimation during training uses both source and target samples. We therefore view these settings as promising extensions rather than claims directly validated in this work.
> ### Q4. Computational Overhead
> Relative to $f$-DAL/$f$-DD, our method uses the same backbone and training pipeline, with no additional encoder, classifier, or discriminator network. In the default affine setting, $\tau(x)=ax-b$ adds only two scalar parameters $(a,b)$; with adaptive divergence selection, we further optimize two scalars $(\beta,\phi)$. The extra computation is limited to pointwise transforms and a few batch-level scalar operations, with no extra feature-extraction pass, prediction head, or retraining over candidate divergences. We will clarify this more explicitly in the revision.

---

### Official Review · Reviewer_4K4q · 2026-03-11

**Soundness:** 3
**Presentation:** 3
**Significance:** 3
**Originality:** 3
**Overall Recommendation:** 4
**Confidence:** 3

**Summary:**

For unsupervised domain adaptation (UDA), where a model trained on labeled source data is expected to generalize to an unlabeled target domain with distribution shift, current existing methods measure the discrepancy between domains using a divergence metric (Jensen-Shannon, KL divergence, Wasserstein distance), which may be suboptimal for different types of domain shifts. Next, during training this results in not learning about the held target domain.

To mitigate this, the authors propose a framework that adaptively learns the divergence measure instead of fixing it beforehand. Their method introduces a tighter variational representation for f-divergences by inserting a learnable, Lipschitz-continuous transformation, leading to more accurate discrepancy estimates across domains. They also use a likelihood-based criterion to adaptively select the divergence family from data. This paper derives new target risk bounds that combine a transformed source risk, a proposed discrepancy term, and an ideal hypothesis error, along with finite-sample guarantees. They validate their method on common domain adaptation benchmarks and demonstrate its effectiveness.

**Compliance With Llm Reviewing Policy:**

Affirmed.

**Final Justification:**

I will maintain my initial score, which already signals acceptance of the paper.

**Key Questions For Authors:**

**Questions**

- For the Learnable transform introduced the authors must provide additional details regarding its implementation
- The authors assume access to the target distribution. In cases where the target distribution consists of mixed distributions or access is not provided during training, how would the method work?
- Could the authors detail the additional training parameters or compute those required during source training?
- Could the author provide more details about Step 11, where the h_cls is updated with gradient ascent? I do not see a reference to the stated equations
- Since recent works in domain adaptation, do not assume access to target set, I recommend the authors to stress this in their work such that there is no ambiguity

**Limitations:**

Yes

**Strengths And Weaknesses:**

**Strengths:**
- The paper’s contribution seems solid due to the learnable transformation, which results in more discrepancy between the domains
- Despite the appendix, the authors also demonstrate with an additional backbone ViT apart from ResNet.

**Weaknesses**
- Statistically, the numbers from Table 1 do not necessarily show that the proposed method performs best. I recommend that authors also bold the baseline methods whose values fall in the same range as theirs
- Why didn't the authors include recent baselines from Domain adaptation, and also standard domain adaptation methods from the ones listed in Domainbed [1].
- The results of extensive evaluation with datasets from domain adaptation, such as Terra Incognita, Domainbed, and where the number of domains is even more than the mid or small datasets used by the paper are missing


References:
1. Gulrajani, Ishaan, and David Lopez-Paz. "In search of lost domain generalization." arXiv preprint arXiv:2007.01434 (2020).

---

> ### Author Rebuttal · Authors · 2026-03-30
>
> We thank the reviewer for the constructive feedback and helpful questions.
> ### W1. Statistical Reporting
> We thank the reviewer for raising this point. Table 1 reports mean ± std over three independent random seeds, and **Appendix I.3.2** describes our statistical analysis protocol and representative significance tests. We agree that the current presentation does not make this sufficiently explicit. In the revision, we will clarify the statistical interpretation and indicate where differences are not statistically significant. Rather than bolding methods based on visually similar values, we believe significance-aware reporting is more principled, since numerical proximity alone does not imply statistical equivalence.
> ### W2/W3/Q2/Q5. Scope and Benchmark Choice
> We thank the reviewer for these questions. Our paper studies standard single-source UDA, where unlabeled target samples are available during training. To address the concern about broader evaluation, we additionally ran a DeiT-S/16 experiment on the pre-specified 12-task DomainNet subset (C/P/R/S). Under the same backbone and protocol, our method achieves the best average accuracy (58.7), outperforming DANN (52.5), $f$-DAL (55.6), and $f$-DD (56.9); see **reviewer feD4 W1&Q3** for details. This suggests that the method remains effective on a larger and more heterogeneous benchmark.
>
> DomainBed and Terra Incognita are valuable benchmarks, but they are primarily used for domain generalization and related multi-domain settings, where target-domain samples are unavailable during training. Our paper instead focuses on standard UDA, so these settings are not directly comparable. We will clarify this in the revision.
>
> More generally, the method applies whenever unlabeled target samples are available during training, including multi-modal target distributions, since discrepancy estimation and adaptive divergence selection operate on the empirical target marginal without assuming unimodality. If no target data are available during training, the current formulation is not directly applicable. Extending it to DG, source-free UDA, or multi-source adaptation would require nontrivial changes to both the objective and the theory, and we view these as important future directions.
>
> ### Q1. Details of Learnable Transform
> We thank the reviewer for this suggestion. We agree that the implementation of $\tau$ should be stated more explicitly. In our implementation, $\tau$ is not an additional neural module, but a simple scalar transform applied pointwise in the transformed source classification term and the tighter-VR discrepancy term. In the main experiments, we use the affine form $\tau(x)=ax-b$, with $(a,b)$ initialized as $(1,0)$ and learned jointly with the network. We also study power, exponential, and sigmoid forms as ablations; their initialization details are given in **Appendix G.2**. We will make this explicit in the revision.
> ### Q3. Additional Parameters and Compute Overhead
> We thank the reviewer for this question. Relative to the $f$-DAL/$f$-DD setup, our method adds no extra encoder, classifier, or selector network. In the affine setting used in the main experiments, the only additional learnable parameters are the two scalars $(a,b)$ in $\tau(x)=ax-b$. With adaptive divergence selection, we further optimize two scalars $(\beta,\phi)$, where $\beta$ specifies the $\beta$-divergence and is related to $\alpha$ by reparameterization.
>
> The overhead is minimal. We do not solve Eq. (11) with a separate inner loop or repeated search over candidate divergences; instead, $(\beta,\phi)$ are updated online together with the main network. The extra cost is limited to pointwise application of $\tau$ and updates of a few scalar variables, which is negligible relative to the encoder forward/backward passes. Inference introduces no additional network overhead. We will clarify this more explicitly in the revision.
> ### Q4. Step 11 in Algorithm 1
> We thank the reviewer for pointing this out. In Algorithm 1, Step 11 updates the auxiliary head $h'\_{\mathrm{cls}}$, not the primary classifier $h\_{\mathrm{cls}}$. This step implements the inner maximization over $h'$ in Eqs. (13) and (14).
>
> Specifically, Step 10 performs gradient descent on
> $$
> L = L_{\mathrm{cls}} + \eta\,\tilde d^{\tau}\_{\hat\mu,\hat\nu}(h,h')
> $$
> with respect to $h_{\mathrm{rep}}, h_{\mathrm{cls}}, \tau$, while Step 11 performs gradient ascent on $\tilde d^{\tau}\_{\hat\mu,\hat\nu}(h,h')$ with respect to $h'\_{\mathrm{cls}}$:
>
> $$
> \theta'\_{\mathrm{cls}}
> \leftarrow
> \theta'\_{\mathrm{cls}}+
> \rho \nabla_{\theta'_{\mathrm{cls}}}
> \tilde d^{\tau}\_{\hat\mu,\hat\nu}(h,h').
> $$
>
> Since $L\_{\mathrm{cls}}$ does not depend on $h'\_{\mathrm{cls}}$, Step 11 affects only the discrepancy term. There is no gradient-ascent update on the primary classifier $h_{\mathrm{cls}}$. We agree that this was not sufficiently clear, and we will revise Algorithm 1 to label $h'_{\mathrm{cls}}$ explicitly and cite Eqs. (13) and (14).

---

> > ### Author Rebuttal · Reviewer_4K4q · 2026-04-03
> >
> > Thanks to the authors for their efforts in answering my questions.

---

> > > ### Author Response · Authors · 2026-04-04
> > >
> > > Thank you very much for your time and for your thoughtful follow-up. We sincerely appreciate your recognition that the concerns have been fully addressed. We are grateful for your insightful questions, which helped us improve the clarity and completeness of the paper. If you feel that the revisions and responses have strengthened the work, we would greatly appreciate it if you could consider reflecting this in your final score.

---

### Official Review · Reviewer_feD4 · 2026-03-12

**Soundness:** 4
**Presentation:** 3
**Significance:** 2
**Originality:** 3
**Overall Recommendation:** 4
**Confidence:** 4

**Summary:**

The paper proposes a tighter variational representation (Tighter-VR) for estimating f-divergence by introducing a learnable transformation inside the Fenchel-duality bound, resulting in a strictly tighter discrepancy estimate for unsupervised domain adaptation. The motivation is that most UDA methods fix a single f-divergence to measure source–target discrepancy, which may be suboptimal under heterogeneous shifts. While classical UDA theory (Ben-David et al.) bounds target risk using an H \Delta H-divergence, many recent methods minimize specific f-divergences. The proposed framework generalizes these approaches, provides a tighter bound, and recovers affine and power transforms as special cases.

The method uses a standard encoder–classifier network where a shared critic aligns source and target representations. The Training objective is a combination of  source classification loss (under the learned transform )  and the Tighter-VR discrepancy term. The paper also derives a target-risk bound with three components- transformed source risk, domain discrepancy, and an ideal-hypothesis residual . They conduct experiments on Office-31, Office-Home, Digits, and VisDA-2017 and show  gains over prior f-divergence methods. The ablations show that both the tighter bound and the adaptive divergence selection are complementary

**Compliance With Llm Reviewing Policy:**

Affirmed.

**Final Justification:**

Upon reading the rebuttal and other reviews. I maintain my positive score and recommend for acceptance.

**Key Questions For Authors:**

1. The paper only compares against other f-divergence methods, despite Wasserstein and optimal transport being natural competitors in UDA. Can the authors include at least one OT-based baseline? If the gains hold up there too, it would
  considerably strengthen the paper.
 2. The Office-Home results are weak and inconsistent across domain pairs. Given the theoretical guarantees, what explains this? Is the bound loose in practice on this benchmark, or is something else going on?
 3. How does the method perform with ViT or CLIP backbones? CLIP already gives strong cross-domain features, so it is not obvious that additional alignment helps. This is important for understanding how relevant the method is to current practice.
 4. Is there a fundamental reason the method cannot extend to open-set or partial DA, or is it just out of scope? I suspect this has to do with the third term but  even a rough answer here would help

**Limitations:**

Yes

**Strengths And Weaknesses:**

Strengths :
The paper is well-written and easy to follow. The theoretical analysis is rigorous, including solid Rademacher complexity–based generalization bounds. The experiments on standard UDA benchmarks align well with the theoretical claims.

Weaknesses:
The experiments are run on fairly old benchmarks (Office-31, Office-Home, Digits, VisDA-2017) with ResNet/LeNet backbones. While this is somewhat acceptable for a theory-focused paper, the improvements are inconsistent for example Office-Home gains are quite small and standard deviations are also not reported. For a paper that leans heavily on theory to justify lower target risk, the empirical story needs to be stronger.   The comparison set is also narrow. The paper discusses Wasserstein and optimal transport methods in related work but never compares against them. Given that these are natural competitors, leaving them out is a notable gap. More broadly, UDA has moved on significantly. Open-set, partial, and multi-source settings are now standard evaluation grounds. We do need new theoretical frameworks for them as well.
Further, the paper does include a ViT experiment, but it is in the appendix and not given serious treatment. With widespread adoption of foundation models like CLIP etc, it is  unclear how the approach performs in problem settings that the community actually cares about right now.

---

> ### Author Rebuttal · Authors · 2026-03-30
>
> We thank the reviewer for the careful reading and constructive comments. We have added new results and clarified the scope and empirical findings accordingly.
>
> ### W1&Q3. ViT on DomainNet
>
> We agree that the original evaluation focused on older benchmarks and CNN backbones. To address this, we added a transformer-backbone evaluation on DomainNet using DeiT-S/16 over the 12-task C/P/R/S subset. All methods follow the same backbone and protocol. For $f$-DD we use the fixed $\chi^2$ divergence, while our method uses adaptive $\beta$-divergence with an affine transform and fixes $\beta$ after 10 epochs.
>
> As shown below, our method achieves the best average accuracy (58.7), outperforming DANN (52.5), $f$-DAL (55.6), and $f$-DD (56.9). It also improves over the strongest baseline on all 12 transfer directions, suggesting that the proposed discrepancy remains effective under larger and more heterogeneous shifts.
>
> | Method | C->P | C->R | C->S | P->C | P->R | P->S | R->C | R->P | R->S | S->C | S->P | S->R | Avg |
> | ---   | --- | --- | --- | --- | --- | --- | --- | --- | --- | --- | --- | --- | --- |
> | DANN  | 48.3 | 52.0 | 59.2 | 47.3 | 62.1 | 46.7 | 65.2 | 63.4 | 62.1 | 46.2 | 36.7 | 40.3 | 52.5 |
> | $f$-DAL | 51.2 | 57.3 | 62.1 | 50.2 | 64.7 | 50.3 | 68.5 | 67.2 | 64.9 | 48.7 | 37.9 | 44.7 | 55.6 |
> | $f$-DD  | 52.7 | 58.2 | 62.7 | 51.7 | 65.4 | 52.2 | 69.7 | 68.8 | 65.7 | 50.2 | 38.6 | 46.3 | 56.9 |
> | Ours  | 54.1 | 60.3 | 64.2 | 53.6 | 67.5 | 54.1 | 71.2 | 70.6 | 68.0 | 51.9 | 40.2 | 48.8 | 58.7 |
>
> A CLIP-based evaluation is also relevant, but it introduces additional design choices beyond the standard closed-set UDA setting studied here. To isolate the core question of whether our discrepancy remains useful with a stronger modern visual backbone under the same protocol, we evaluate DeiT-S/16 instead.
>
> ### W2&Q1. OT-based Baseline
>
> We agree that limiting the comparison to prior $f$-Divergence methods leaves out an important cross-family reference, since OT/Wasserstein methods are natural competitors in discrepancy-based UDA. We therefore include representative OT-based baselines below:
>
> | Method | Office-31 | Office-Home | VisDA-2017 |
> | --- | --- | --- | --- |
> | JDOT [1] | 83.5 | 60.5 | 74.7 |
> | DeepJDOT [2] | 86.2 | 50.6 | 77.4 |
> | MLOT [3] | 87.8 | 66.2 | 73.0 |
> | Ours | 90.4 | 69.9 | 78.9 |
>
> These comparisons show that our method is competitive not only against prior $f$-Divergence baselines but also against representative OT-based approaches. Our claim is not that $f$-Divergence methods universally dominate OT methods, but that a tighter and adaptive discrepancy can provide a stronger alignment signal than a fixed pre-selected discrepancy.
>
> ### W3&Q2. Office-Home gains and missing standard deviations
> We agree that the gains on Office-Home are smaller and less uniform than on VisDA. This likely reflects the fact that VisDA is larger and more challenging, with a stronger domain shift and thus more room for improvement under the standard ResNet-101 protocol. Office-Home offers less headroom, so even a better discrepancy surrogate may produce only modest absolute gains. It is also more heterogeneous across domain pairs, with greater variation in transfer difficulty and optimization sensitivity, leading to less uniform improvements.
>
> This does not contradict the theory. Our result is meant to justify the surrogate/reduction and provide a principled target-risk decomposition, not to tightly predict the gain on every transfer pair. The bound may be numerically loose on this benchmark while still offering useful qualitative justification.
>
> We also agree that the statistical reporting should be clearer. Due to space constraints, we do not include the full mean ± std table here, but the Office-Home average over domain pairs is 69.9 ± 0.3; we will report mean ± std for all transfer pairs in the revision.
>
>
>
>
> ### Q4&W4. Extension to open-set or partial DA
>
> We agree that these DA variants are important, and we will clarify the scope more carefully. The current paper focuses on standard closed-set UDA.
>
> For partial DA, our transformed discrepancy and adaptive divergence selection could in principle be integrated into existing pipelines, but additional source-class reweighting or filtering would still be needed to handle source-private classes. For open-set DA, the current method is not a complete solution as is, since explicit unknown-class handling is required. For source-free DA, the present formulation is also not directly applicable, because discrepancy estimation during training uses both source and target samples.
>
> We therefore view these settings as promising extensions rather than claims directly validated in this work.
>
> **Reference**
>
> [1] Joint distribution optimal transportation for domain adaptation. NeurIPS 2017.
>
> [2] DeepJDOT: Deep Joint Distribution Optimal Transport for Unsupervised Domain Adaptation. ECCV 2018.
>
> [3] Metric Learning in Optimal Transport for Domain Adaptation. IJCAI 2020.

---

> > ### Author Rebuttal · Reviewer_feD4 · 2026-04-03
> >
> > I thank the authors for their detailed response. Adding the OT and DeiT experiments have strengthened the paper.
> > I also read the other reviews and their corresponding responses.

---

> > > ### Author Response · Authors · 2026-04-04
> > >
> > > Thank you very much for your thoughtful follow-up and for taking the time to carefully consider our rebuttal. We are particularly grateful for your suggestions on including optimal transport baselines and DeiT-based experiments, which have helped us further strengthen the paper. If you feel that the revised version better reflects the contributions and empirical support of our work, we would be grateful if you would consider updating the score. Thank you again for your valuable feedback.

---

### Decision · Program_Chairs · 2026-04-30

**Decision:**

Accept (regular)

**Comment:**

This paper presents a framework for unsupervised domain adaptation (UDA) that adaptively selects an $f$-divergence family and tightens its variational lower bound via a learnable Lipschitz transform. All reviewers leaned positive, highlighting the sound theoretical foundations and the adaptive discrepancy mechanism as primary strengths. Initial concerns regarding the reliance on older empirical baselines were adequately addressed during the rebuttal through additional optimal transport comparisons and transformer-based evaluations, though reviewers noted the method's scope is inherently limited to closed-set UDA. I thus recommend acceptance. If the paper is accepted, the authors should incorporate evaluation results provided during the rebuttal in the final paper.